# Spermatocytes have the capacity to segregate chromosomes despite centriole duplication failure

Marnie W Skinner[1,2], Carter J Simington[1], Pablo López-Jiménez [3,6], Kerstin A Baran[1], Jingwen Xu [1], Yaron Dayani[2,4], Marina V Pryzhkova[2,4], Jesús Page[3], Rocío Gómez [3], Andrew J Holland [5] & Philip W Jordan [1,2✉]

## Abstract

**Centrosomes are the canonical microtubule organizing centers (MTOCs) of most mammalian cells, including spermatocytes. Centrosomes comprise a centriole pair within a structurally ordered and dynamic pericentriolar matrix (PCM). Unlike in mitosis, where centrioles duplicate once per cycle, centrioles undergo two rounds of duplication during spermatogenesis. The first duplication is during early meiotic prophase I, and the second is during interkinesis. Using mouse mutants and chemical inhibition, we have blocked centriole duplication during spermatogenesis and determined that non-centrosomal MTOCs (ncMTOCs) can mediate chromosome segregation. This mechanism is different from the acentriolar MTOCs that form bipolar spindles in oocytes, which require PCM components, including gamma-tubulin and CEP192. From an in-depth analysis, we identified six microtubule-associated proteins, TPX2, KIF11, NuMA, and CAMSAP1-3, that localized to the non-centrosomal MTOC. These factors contribute to a mechanism that ensures bipolar MTOC formation and chromosome segregation during spermatogenesis when centriole duplication fails. However, despite the successful completion of meiosis and round spermatid formation, centriole inheritance and PLK4 function are required for normal spermiogenesis and flagella assembly, which are critical to ensure fertility.**

**Keywords** Centriole; Centrosome; Meiosis; Spermatogenesis; PLK4
**Subject Category** Cell Cycle

## Introduction

Infertility remains a public health issue that affects around 15% of couples worldwide (Sun et al, 2019). A major cause of infertility can arise from errors during meiosis (Potapova and Gorbsky, 2017). Meiotic cells undergo one round of DNA replication, followed by two successive rounds of chromosome segregation, termed meiosis I and meiosis II. In meiosis I, homologous chromosome pairs, which are linked via crossover recombination events, are segregated. The sister chromatid pairs, which had remained associated with one another during meiosis I, segregate during meiosis II. The accurate separation of the chromosomes via bipolar spindles during meiosis is critical, as aneuploidy events can lead to infertility, spontaneous abortion, and the development of genetic disease in the resulting offspring (Potapova and Gorbsky, 2017). The mechanism of bipolar spindle formation is sexually dimorphic in mammals. In oocytes, bipolar spindles are formed via an acentriolar microtubule organizing center (aMTOC), whereas spermatocytes rely on a canonical centriole-containing centrosome for bipolar spindle formation (Lerit and Poulton, 2016; Gruhn and Hoffmann, 2022; So et al, 2019; Schatten and Sun, 2009). While the importance of the centrosome is well established in mitotic cells, very little is known about the regulation of centrosome biogenesis during spermatogenesis.

A centrosome consists of two centrioles and associated proteins known as the pericentriolar matrix (PCM). Centrioles are cylindrical complexes made up of nine triplicate microtubules and exist in pairs orthogonally oriented to each other (Jana, 2021). The older centriole, termed the mature parent centriole, has additional maturation markers known as subdistal and distal appendages that are required for the recruitment of PCM to the centrioles and centriole membrane docking during ciliogenesis, respectively (Tanos et al, 2013; Mazo et al, 2016). During mitosis and meiosis, the PCM acts as the MTOC by concentrating γ-tubulin ring complexes (γ-TuRC) that serve as nucleation sites for the assembly of bipolar spindles required for chromosome segregation (Bornens, 2002). Following meiosis II, haploid spermatids undergo spermiogenesis, where the centrioles become the foundation for the microtubule structure responsible for flagella formation (Avidor-Reiss et al, 2020; Bettencourt-Dias et al, 2005).

Polo-like kinase 4 (PLK4) is considered the "master regulator" of centriole duplication during interphase in mitotically dividing cells (de Cárcer et al, 2011; Zitouni et al, 2014). PLK4-mediated centriole duplication in mitotic cells is tightly regulated to ensure this duplication event occurs only once per cell cycle (Bettencourt-Dias et al, 2005;

[1]Department of Biochemistry and Molecular Biology, Johns Hopkins University Bloomberg School of Public Health, Baltimore, MD, USA. [2]Department of Biochemistry and Molecular Biology, Uniformed Services University of the Health Sciences, Bethesda, MD, USA. [3]Department of Biology, Autonomous University of Madrid, Madrid, Spain. [4]The Henry M. Jackson Foundation for the Advancement of Military Medicine, Bethesda, MD, USA. [5]Department of Molecular Biology and Genetics, Johns Hopkins University School of Medicine, Baltimore, MD, USA. [6]Present address: MRC Laboratory of Medical Sciences, London W12 0NN, UK. ✉E-mail: philip.jordan@usuhs.edu

Habedanck et al, 2005; Slevin et al, 2012; de Cárcer et al, 2011; Sillibourne and Bornens, 2010; Klebba et al, 2013, 2015; Cunha-Ferreira et al, 2013; Park et al, 2019). Two PLK4 proteins dimerize via crypto polo box interactions and activate each other via trans-autophosphorylation of their downstream regulatory element domain during the G1 phase (Klebba et al, 2015). The active PLK4 dimer induces centriole duplication during the G1 to S phase transition, and PLK4 then undergoes proteasomal degradation (Nigg and Raff, 2009; Rogers et al, 2009; Hatch et al, 2010). This self-degradation mechanism of PLK4 helps to prevent overduplication of centrioles. PLK4 misregulation is known to promote tumorigenesis and is associated with multiple hallmarks of cancer, such as resisting cell death, genome instability and mutations, and sustained proliferative signaling (Zhang et al, 2021; Wang et al, 2019; Ko et al, 2005; Holland and Cleveland, 2014)

Orthologs of PLK4 have been previously assessed during meiosis in both *Drosophila melanogaster* and *Caenorhabditis elegans*. In these model organisms, PLK4 has been found to be important for centriole duplication during spermatogenesis (Peters et al, 2010; Bettencourt-Dias et al, 2005). However, the function of PLK4 during mammalian spermatogenesis had not been determined. Centriole duplication during mammalian meiosis differs markedly from mitosis in that there are two centriole duplication events that are not linked to the G1 to S phase transition (Wellard et al, 2021; Alfaro et al, 2021). Instead, centrioles are duplicated during early prophase (leptotene stage) of meiosis I and again in interkinesis before entering meiosis II (Wellard et al, 2021; Alfaro et al, 2021). Herein, we report the localization pattern of PLK4 during mouse and human spermatogenesis, and we demonstrate the importance of PLK4 regarding centriole duplication and maturation, centrosome biogenesis, and spermiogenesis. We also show that if centriole duplication is prevented, mouse spermatocytes can still proceed through meiosis I and II and form haploid spermatids. This relies on spermatocytes forming non-centrosomal MTOCs (ncMTOCs) that mediate bipolar spindle formation.

## Results

### PLK4 localizes to the centrosome throughout mammalian spermatogenesis

In mitotically dividing cells, PLK4 protein levels are tightly regulated to ensure centriole duplication only occurs at the G1 to S transition (Nigg and Holland, 2018). While the detailed mechanics behind PLK4 activation at the G1 to S transition have not been fully elucidated, this regulation is mediated by PLK4 dimerization and trans-autophosphorylation, which triggers PLK4 polyubiquitination and subsequent proteolysis (Cunha-Ferreira et al, 2013; Holland et al, 2010; Klebba et al, 2013; Holland et al, 2012; Guderian et al, 2010). As a result, it is challenging to visualize PLK4 via western blot analysis (Byrne et al, 2020). In contrast, PLK4 was reported to be detected via western blot throughout the first wave of spermatogenesis, suggesting higher levels of PLK4 are needed during spermatogenesis compared to mitotically dividing cells (Jordan et al, 2012). We performed a series of immunofluorescence (IF) microscopy approaches to assess the localization pattern of PLK4 during spermatogenesis. Immunolabelled cryosections of seminiferous tubules from adult wild-type (WT) mice demonstrated that PLK4 is present as two foci during meiotic

prophase that colocalized with centrosomal marker CEP192 (Fig. 1A,B). PLK4 signal remained present following meiotic divisions in round spermatids and during all stages of spermiogenesis (Fig. 1A,B). To test the conservation of the PLK4 localization pattern, we immunolabelled cryosections of human testes with an antibody against PLK4. PLK4 colocalized with the centrosome component γ-tubulin in primary spermatocytes (Fig. 1C). Furthermore, PLK4 localization was observed as two foci in SYCP3-positive human primary spermatocytes (Fig. 1E). As observed in mouse seminiferous tubules, PLK4 remained present in round and elongating spermatids (Fig. 1D,F).

To closely assess the localization of PLK4 in spermatocytes at different stages, we took advantage of the first semi-synchronous wave of spermatogenesis (Fig. 1G,H). In prior work, we established that the first of two rounds of centriole duplication occurs during the leptotene sub-stage of meiotic prophase (Wellard et al, 2021; Alfaro et al, 2021). We determined that PLK4 localizes to both parent centrioles prior to centriole duplication (Fig. 1G,H). Following the first round of centriole duplication, PLK4 localizes in between the parent centrioles of each centriole pair. When centrosomes separate during late prophase, PLK4 continues to be detected between centriole pairs, and this remains the case during the first meiotic division. Following meiosis I, centrioles duplicate again at interkinesis, and PLK4 signal persists between the centriole pairs during meiosis II and in the resulting round spermatids (Fig. 1G,H). During the transformation of round spermatids towards spermatozoa (spermiogenesis), PLK4 localization broadens around the centrioles at the base of the spermatid, where the flagella will form (Fig. 1G,H). In addition, PLK4 signal can be detected as a ring that surrounds the spermatid nucleus (Fig. 1G,H). Collectively, this localization pattern is reminiscent of the manchette and perinuclear ring structures, respectively, which are important for the elongation of the sperm nucleus and the formation of functional spermatozoa (Lehti and Sironen, 2016; Kato et al, 2004). PLK4 was also observed as a ring-like structure and at the manchette in elongating human spermatids (Fig. 1D,F). These observations indicate that PLK4 localization patterns during spermatogenesis are conserved between mice and humans.

Taken together, we demonstrate that unlike what has been described for mitotically dividing cells, PLK4 remains present at the centrosome following both rounds of centriole duplication during meiosis. In addition, PLK4 localization during spermiogenesis mimics components of the perinuclear ring and manchette, indicating that PLK4 has roles during spermatid to spermatozoa morphogenesis.

### Overexpression of PLK4 in primary spermatocytes leads to centriole overduplication in meiosis I

If PLK4 is overexpressed or not degraded prior to the next mitotic cycle, centrioles undergo amplification, which can result in the formation of multipolar spindles that cause chromosome missegregation (Moyer et al, 2015; Holland et al, 2012). Therefore, it was surprising to see that PLK4 remains associated with the centrioles throughout meiosis, especially considering that centriole duplication occurs at two defined times (Fig. 1F) (Wellard et al, 2021; Alfaro et al, 2021). We wondered whether the function of PLK4 in spermatocytes was negatively regulated following centriole

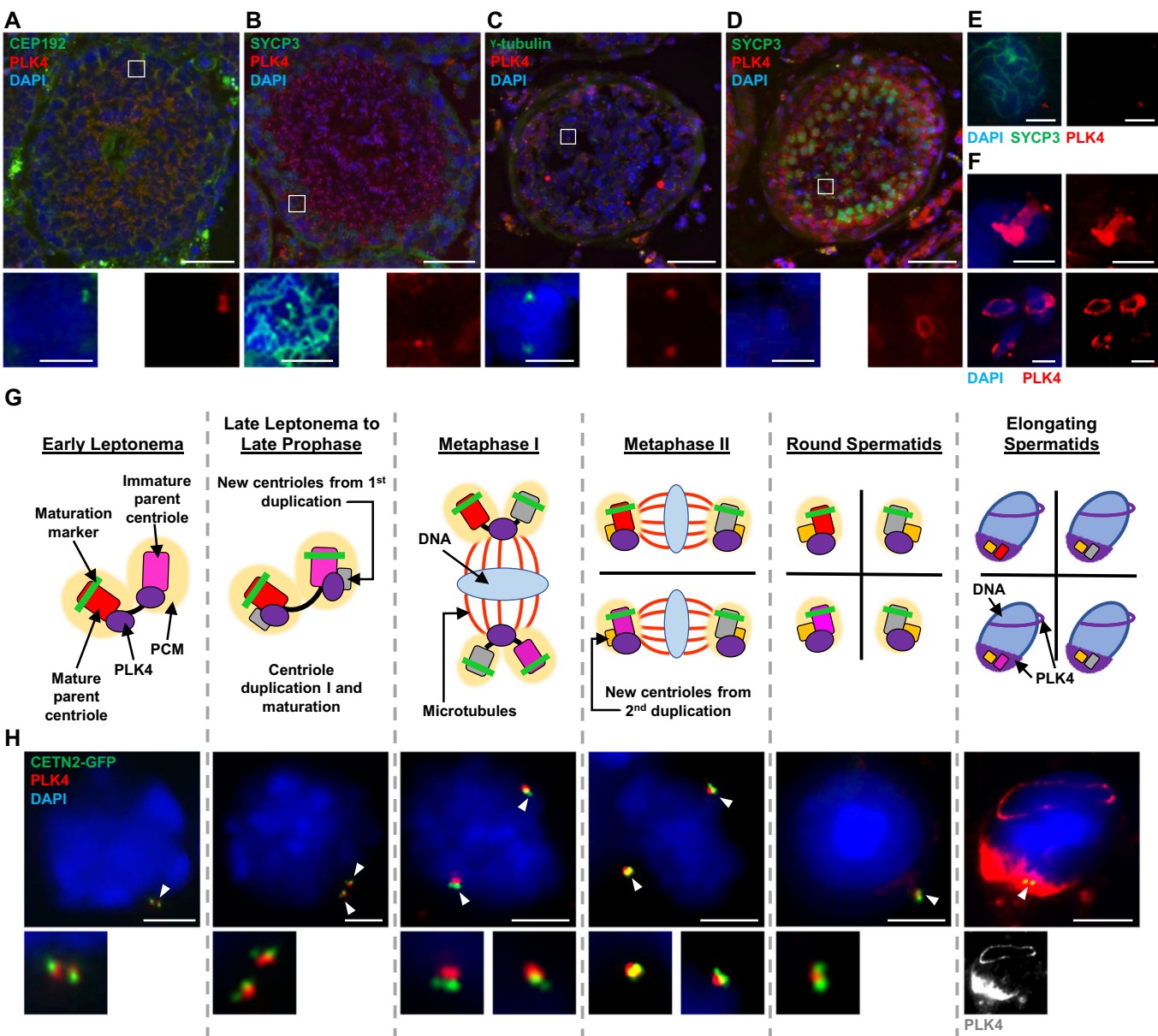

**Figure 1. Assessment of PLK4 localization in murine and human spermatocytes throughout spermatogenesis.**

(**A**) Cryosection (16 μm thick) of an adult WT (26 weeks) mouse testis immunolabeled against CEP192 (green) and PLK4 (red) and stained with DAPI (blue). Scale bar = 50 μm. Zoomed images of PLK4 localization with CEP192 are shown directly below. Scale bar = 5 μm. (**B**) Cryosection (16 μm thick) of an adult WT (26 weeks) mouse testis immunolabeled against SYCP3 (green) and PLK4 (red) and stained with DAPI (blue). Scale bar = 50 μm. Zoomed images of PLK4 localization in SYCP3-positive cells are shown directly below. Scale bar = 5 μm. (**C**) Cryosection (16 μm thick) of an adult (24 years) human testis immunolabeled against γ-tubulin (green) and PLK4 (red) and stained with DAPI (blue). Scale bar = 50 μm. Zoomed images of γ-tubulin and PLK4 co-localization are shown directly below. Scale bar = 5 μm. (**D**) Cryosection (16 μm thick) of an adult (24 years) human testis immunolabeled against SYCP3 (green) and PLK4 (red) and stained with DAPI (blue). Scale bar = 50 μm. Zoomed images of PLK4 localization at the perinuclear ring are shown directly below. Scale bar = 5 μm. (**E**) Cryosection (16 μm thick) of an adult (24 years) human testis immunolabeled against SYCP3 (green) and PLK4 (red) and stained with DAPI (blue). Images demonstrate PLK4 localization in prophase I spermatocytes. Scale bars = 5 μm. (**F**) Cryosections (16 μm thick) of an adult (24 years) human testis immunolabeled against PLK4 (red) and stained with DAPI (blue). Images demonstrate PLK4 localization in elongating spermatids. Scale bars = 5 μm. (**G**) Diagram of the PLK4 localization pattern during spermatogenesis aligned with cells from the corresponding stages of spermatogenesis captured from tubule squash preparations performed on WT control mice (**H**). In the diagram, red rectangle with rounded corners = mature parent centriole, pink rectangle with rounded corners = immature parent centriole, green bar = maturation marker, the yellow oval = PCM, purple shapes = PLK4, gray rectangle with rounded corners = new centrioles from first centriole duplication, yellow rectangle with rounded corners = new centriole from second centriole duplication, blue oval = DNA, red lines = microtubules. (**H**) Control spermatocytes harboring CETN2-GFP (green) were immunolabeled against PLK4 (red) and stained with DAPI (blue). The white arrowheads indicate the centrosome. Zoomed images of the centrioles (or PLK4 staining at the elongating spermatid stage) are shown directly below the corresponding image. Scale bars = 5 μm.

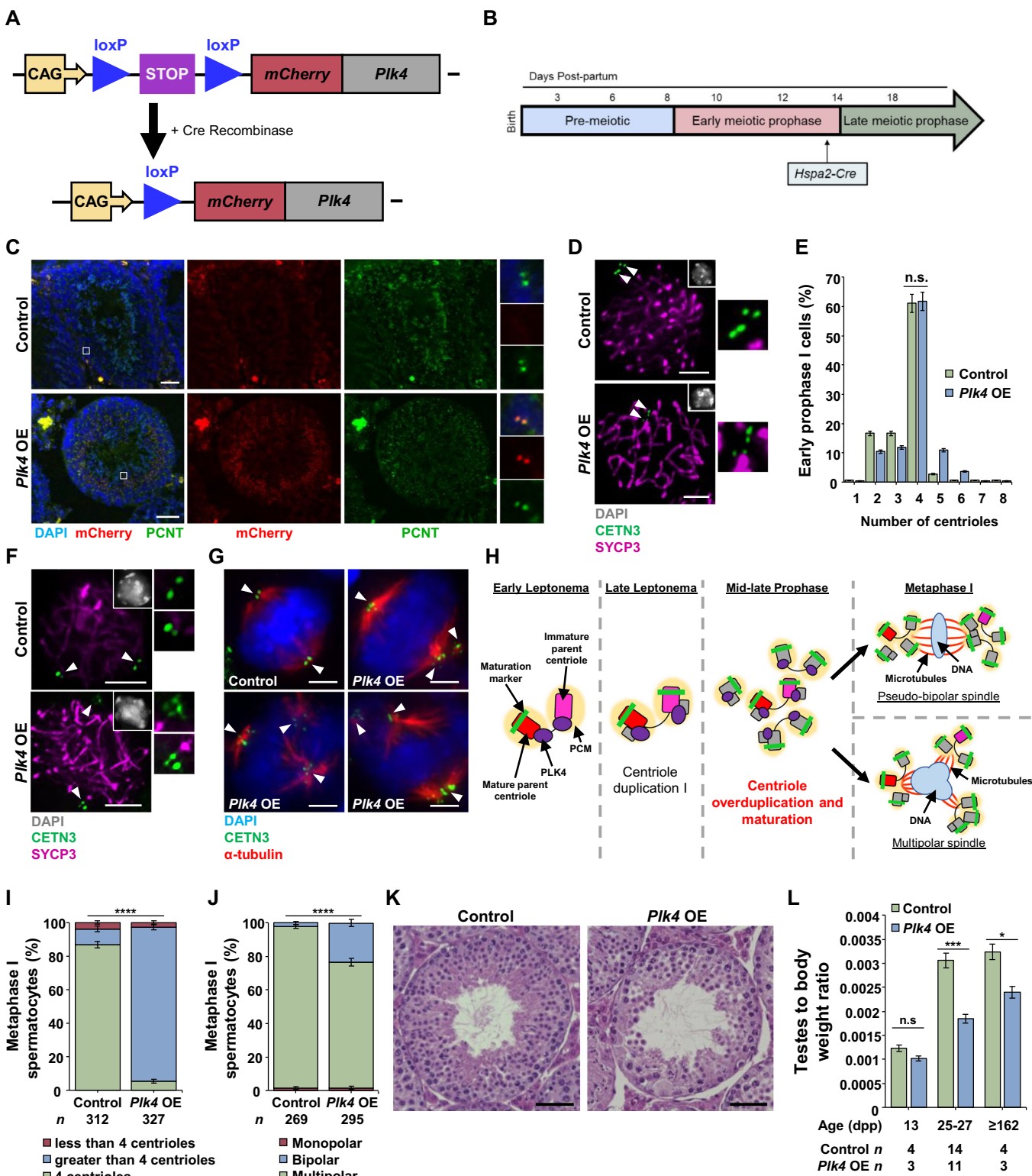

duplication and if we could override this by stimulating the overexpression of PLK4. We accomplished this using a conditional *Plk4* overexpression (OE) mouse model (Fig. 2A,B; see "Methods") (Marthiens et al, 2013). An mCherry-*Plk4* transgene driven by a CAG promoter was conditionally expressed when a stop cassette,

flanked by loxP sites, was removed via Cre-mediated recombination (Fig. 2A). To stimulate overexpression following the first centriole duplication, we used the *Hspa2* promoter-driven Cre recombinase, which is expressed during mid-prophase I in spermatocytes (Fig. 2B) (Inselman et al, 2010).

 

**Figure 2. PLK4 OE leads to centriole overduplication by late prophase in mouse spermatocytes.**

(A) Diagram of the *Plk4* OE allele before and after Cre-mediated recombination. The non-excised allele harbors the CAG promoter (yellow box and arrow) upstream of two loxP sites (blue triangle), flanking a stop cassette (purple box). Further downstream is a non-endogenous copy of *Plk4* fused with an mCherry tag (red and gray boxes). The stop codon was excised via Cre recombination, leaving a single loxP site and resulting in *Plk4* OE. (B) The *Hspa2* promoter used to drive Cre recombinase expression in the *Plk4* OE mouse model is expressed in mid-prophase I at ~14 dpp. (C) Cryosections (16 μm thick) of an adult WT and *Plk4* OE (26 weeks) mouse testis immunolabeled against PCNT (green) and mCherry (red) and stained with DAPI (blue). Zoomed images of mCherry and PCNT localization are shown to the right of their respective image. Scale bars = 50 μm. (D) Representative images of mid-prophase spermatocytes in control and *Plk4* OE mice immunolabeled against SYCP3 (purple) and CETN3 (green) and stained with DAPI (gray inset in the upper right corner). The white arrowheads indicate the centrosome. Zoomed images of the centrioles are shown to the right of each image. Scale bars = 5 μm. (E) Quantification of CETN3 foci observed during early prophase I in both control and *Plk4* OE spermatocytes. Immunolabeling was performed on three biological replicates with ≥36 spermatocytes quantified per replicate. The total number of cells quantified for control and Plk4 OE mice was 144 and 220, respectively. (F) Representative images of late prophase spermatocytes in control and *Plk4* OE mice immunolabeled against SYCP3 (purple) and CETN3 (green) and stained with DAPI (gray inset in the upper right corner). The white arrowheads indicate the centrosome. Zoomed images of the centrioles are shown to the right of each image. Scale bars = 5 μm. (G) Representative images of metaphase I spermatocytes in control and *Plk4* OE mice immunolabeled against α-tubulin (red) and CETN3 (green) and stained with DAPI (blue). The white arrowheads indicate a centrosome. Scale bars = 5 μm. (H) Diagram of the effect of *Plk4* OE on centrioles during mammalian spermatogenesis. *Plk4* OE spermatocytes underwent centriole overduplication in mid-to-late prophase, which led to pseudobipolar or multipolar spindle formation at metaphase I. Red rectangle with rounded corners = mature parent centriole, pink rectangle with rounded corners = immature parent centriole, green bar = maturation marker, yellow oval = PCM, purple oval = PLK4, gray rectangle with rounded corners = new centrioles from centriole duplication events, blue oval = DNA, red lines = microtubules. (I) Quantification of CETN3 foci per metaphase I spermatocyte. Immunolabeling was performed on three biological replicates with ≥23 spermatocytes quantified per replicate. The total number of cells quantified for control and *Plk4* OE mice was 312 and 327, respectively. *P* value = <0.0001. (J) Quantification of the spindle orientation in metaphase I spermatocytes. Immunolabeling was performed on five biological replicates, with ≥19 spermatocytes quantified per replicate. The total number of cells quantified for control and *Plk4* OE mice was 269 and 295, respectively. *P* value = <0.0001. (K) H&E staining of 5 μm thick testis sections of 95 dpp control or *Plk4* OE mice. Scale bar = 50 μm. (L) Quantification of the average testis to body weight ratio of control and *Plk4* OE mice. Measurements were performed using ≥3 mice for each age group. *P* values for 13 dpp, 25–27 dpp, and ≥162 dpp were 0.0751, 0.0009, and 0.0252, respectively. Data information: For all graphs (E, I, J, L), error bars show mean ± standard error of the mean (SEM). *P* values were obtained from two-tailed Student's *t* test. n.s. (not significant), *$P < 0.05$ **$P < 0.01$, ***$P < 0.0001$, ****$P < 0.0001$.

Through immunohistochemistry of cryosectioned seminiferous tubules, we demonstrated that mCherry-PLK4 localized to the centrosome in spermatocytes (Fig. 2C), which complemented our observations of PLK4 localization in WT spermatocytes (Fig. 1A,B). From the assessment of centriole duplication during prophase I, we observed that the first round of centriole duplication was unaffected in the *Plk4* OE spermatocytes, which was expected given that this process occurs before *Hspa2*-Cre recombinase expression (Fig. 2D,E) (Inselman et al, 2010). However, by mid-prophase I, after *Hspa2*-Cre recombinase expression occurs, centriole overduplication was observed in *Plk4* OE spermatocytes (Fig. 2F). The majority of spermatocytes that progressed to metaphase I exhibited centriole overduplication, which was a rare occurrence to be observed in control metaphase I spermatocytes (Fig. 2G–I). Centriole numbers in *Plk4* OE metaphase I spermatocytes ranged from 5 up to 29. Of these *Plk4* OE metaphase I-staged spermatocytes, 23% contained multipolar spindles compared to 2% of spermatocytes with multipolar spindles observed in controls (Fig. 2J). From our analysis of histological preparations of adult *Plk4* OE seminiferous tubule cross sections, we observed that the aberrancies in centriole number, spindle assembly, and chromosome alignment led to defects in spermatogenesis and germ cell depletion (Fig. 2K). Consequently, *Plk4* OE mice had reduced testis to body weight ratios compared to controls (Fig. 2L). Furthermore, fertility tests demonstrated that *Plk4* OE male mice were infertile (no litters were produced when four *Plk4* OE males were each bred to two fertile WT females for a minimum of 8 weeks).

Taken together, these observations highlight several key points related to PLK4 function and regulation during male meiosis. First, PLK4 is a crucial regulator of centriole duplication during spermatogenesis, as *Plk4* OE leads to centriole overduplication. Second, an unknown regulatory mechanism prevents PLK4 from triggering centriole overduplication in WT spermatocytes, which is overcome by an increase in PLK4 levels in the *Plk4* OE model.

Lastly, an excess of PLK4 alone was sufficient to induce extreme centriole overduplication without the need to increase the levels of other centriole components. This indicates that not only is PLK4 responsible for centriole duplication, but it is also the limiting factor of centriole duplication during spermatogenesis, which is also the case for mitotically dividing cells (Moyer et al, 2015; Holland et al, 2012).

## Inhibition of PLK4 leads to centriole duplication failure

Having demonstrated that PLK4 overexpression during spermatogenesis promotes centriole amplification, we wanted to further examine the role of PLK4 by abrogating its function. It is known that the kinase activity of PLK4 is critical for centriole duplication in mitotically dividing cells (Holland and Cleveland, 2014; Holland et al, 2010). Therefore, we used a PLK4-specific kinase inhibitor, centrinone-B (Wong et al, 2015), on organotypic cultures of mouse seminiferous tubules to determine whether PLK4 kinase activity is required for the first meiotic centriole duplication event at leptonema. In somatic cell lines, the optimal, efficient concentration used was 500 nM (Tkach et al, 2022). However, higher concentrations of centrinone-B were required in our experiments because inhibition of PLK4 relied on media diffusion into fragments of seminiferous tubules that were embedded in agarose gel partially submerged in culture media (Sato et al, 2011; Alfaro et al, 2021). We tested increasing concentrations of centrinone-B from 10 to 100 μM for 8 and 24 h, first assessing cell viability (Fig. 3A–C). Cell viability was predominantly affected after 8 h of culture with 100 μM of centrinone-B, which was the highest concentration assessed (Fig. 3B). Next, we assessed centriole duplication using concentrations that did not significantly affect spermatocyte viability (0, 10, 20, and 50 μM) following 8 and 24 h of culture. Chromosome synapsis, interpreted by SYCP1 and SYCP3 localization, was used to determine spermatocyte prophase

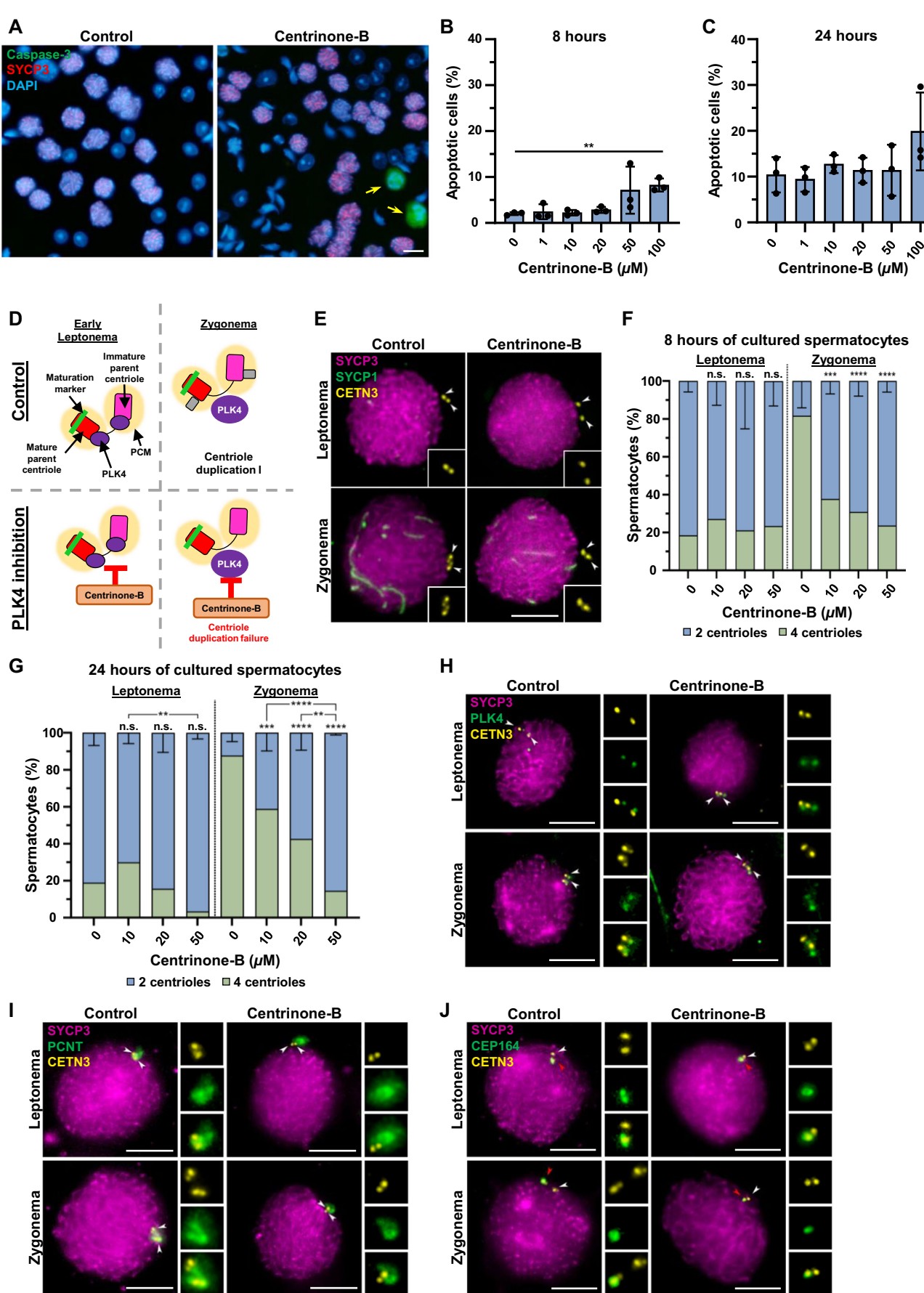

    This is a U.S. Government work and not copyright protection in the US; foreign copyright protection may apply

**Figure 3. PLK4 inhibition leads to centriole duplication failure in vitro.**

(A) Representative images of control and centrinone-B-treated mouse spermatocytes immunolabelled against caspase-3 (green) and SYCP3 (red) and stained with DAPI (blue). Centrinone-B was added to spermatocytes for 8 h. The yellow arrows indicate apoptotic spermatocytes. Scale bars = 10 μm. (B) Quantification of apoptotic spermatocytes in cultures of seminiferous tubules after treatment of vehicle (control) or 10, 20, 50, or 100 μM of centrinone-B for 8 h. The percentage of apoptotic cells was calculated by counting spermatocytes, excluding Sertoli cells and spermatids. Data represents the average percentage of apoptotic spermatocytes for three biological replicates. $P$ values for apoptotic cells treated for 8 h with 10, 20, 50, or 100 μM of centrinone-B compared to the control were 0.6966, 0.6686, 0.1256, 0.1602, and 0.0018, respectively. $P$ values for apoptotic cells treated for 8 h with 10, 20, 50, or 100 μM of centrinone-B compared to the control were 0.7347, 0.4042, 0.7382, 0.0814, and 0.1539, respectively. (C) Quantification of apoptotic spermatocytes in cultures of seminiferous tubules after treatment of vehicle (control) or 10, 20, 50, or 100 μM of centrinone-B for 24 h. The percentage of apoptotic cells was calculated by counting spermatocytes, excluding Sertoli cells and spermatids. Data represents the average percentage of apoptotic spermatocytes for three biological replicates. (D) Diagram illustrating how the inhibition of PLK4 in vitro by centrinone-B leads to centriole duplication failure during early prophase. Red rectangle with rounded corners = mature parent centriole, pink rectangle with rounded corners = immature parent centriole, green bar = maturation marker, yellow oval = PCM, purple oval = PLK4, gray rectangle with rounded corners = new centrioles from first centriole duplication, orange box = centrinone-B, red T shape = indicates inhibition. (E) Representative images of mouse spermatocytes under control and 50 μM centrinone-B treatment conditions for 24 h. Spermatocytes were immunolabeled against SYCP1 (green), SYCP3 (magenta), and CETN3 (yellow). The white arrowheads indicate the centrioles. Zoomed images of the centrioles are inset on each corresponding image. Scale bar = 10 μm. (F) Quantification of CETN3 foci observed during leptonema and zygonema in spermatocytes under control and centrinone-B treatment conditions. Immunolabeling was conducted in three different biological replicates and with increasing concentrations of centrinone-B (10, 20, and 50 μM) for 8 h. $P$ values for leptotene stage spermatocytes treated for 8 h with 10, 20, and 50 μM of centrinone-B compared to the control were 0.909, 0.9969, and 0.9802, respectively. $P$ values for zygotene stage spermatocytes treated for 8 h with 10, 20, and 50 μM of centrinone-B compared to the control were 0.0002, <0.0001, and <0.0001, respectively. (G) Quantification of CETN3 foci observed during leptonema and zygonema in spermatocytes under control and centrinone-B treatment conditions. Immunolabeling was conducted in three different biological replicates and with increasing concentrations of centrinone-B (10, 20, and 50 μM) for 24 h. $P$ values for leptotene stage spermatocytes treated for 24 h with 10, 20, and 50 μM of centrinone-B compared to the control were 0.2775, 0.9424, and 0.0753, respectively. $P$ values for zygotene stage spermatocytes treated for 24 h with 10, 20, and 50 μM of centrinone-B compared to the control were 0.0639, <0.0001, and 0.0012, respectively. (H) Representative images of mouse spermatocytes under control and 50 μM centrinone-B treatment conditions for 24 h. Spermatocytes were immunolabeled against PLK4 (green), SYCP3 (magenta), and CETN3 (yellow). The white arrowheads indicate the centrioles. Zoomed images of the centrioles are shown to the right of each corresponding image. Scale bars = 10 μm. (I) Representative image of mouse spermatocytes under control and 50 μM centrinone-B treatment conditions for 24 h. Spermatocytes were immunolabeled against PCNT (green), SYCP3 (magenta), and CETN3 (yellow). The white arrowheads indicate the centrioles. Zoomed images of the centrioles are shown to the right of each corresponding image. Scale bars = 10 μm. (J) Representative images of mouse spermatocytes under control and 50 μM centrinone-B treatment conditions for 24 h. Spermatocytes were immunolabeled against CEP164 (green), SYCP3 (magenta), and CETN3 (yellow). The white arrowheads indicate the centrioles. The red arrowheads indicate the mature centrioles. Zoomed images of the centrioles are shown to the right of each corresponding image. Scale bars = 10 μm. Data information: For all graphs (B, C, F, G), bars and error bars represent mean ± standard deviation (SD). For graphs (B, C), $P$ values were obtained from one-way ANOVA with Holm–Sidak's comparisons test. For graphs (F, G), $P$ values were obtained from two-way ANOVA with Turkey's multiple comparisons test. n.s. (not significant), **$P < 0.01$, ***$P < 0.001$, ****$P < 0.0001$.

stages. We focused on centriole duplication during the first meiotic division, which occurs prior to the leptotene-to-zygotene transition (Wellard et al, 2021; Alfaro et al, 2021). Results revealed a significant reduction in the number of zygotene stage cells that successfully duplicated their centrioles after 8 or 24 h of 10–50 μM centrinone-B treatments (Fig. 3D–G). Moreover, the number of cells with non-duplicated centrioles increased in higher centrinone-B concentrations (Fig. 3F,G). As the 24-h treatment of 50 μM centrinone-B was the most effective for inhibiting centriole duplication in spermatocytes, all the subsequent analyses were performed using this culture condition.

Next, we assessed PLK4 localization to address whether centrinone-B treatment affected PLK4 distribution and its kinase function. We hypothesized that PLK4 localization would not be affected based on observations previously reported for somatic cells (Wong et al, 2015). For this, we analyzed PLK4, synaptonemal complex formation (SYCP3), and centrioles (CENT3) and determined that PLK4 was detected as a small signal near each centriole (Fig. 3H). This PLK4 distribution was identical regardless of whether centriole duplication had occurred or not in both untreated and centrinone-B-treated spermatocytes. Furthermore, the PLK4 localization pattern observed in these assays corroborates with the localization data shown in Fig. 1H.

Because centrinone-B treatment inhibited centriole duplication, we then asked whether this could also alter centrosome integrity. We did not observe any alterations in the PCM revealed by pericentrin (PCNT) immunostaining in treated spermatocytes compared to the control (Fig. 3I). In addition, we studied

CEP164, a marker for the mature parent centriole in mouse spermatocytes (Wellard et al, 2021; López-Jiménez et al, 2022). No alteration of CEP164 distribution was observed, as the protein was located at the mature parent centriole before and after centriole duplication in control and with unduplicated centrioles following centrinone-B treatment (Fig. 3J).

Collectively, these data demonstrated that PLK4 is the master regulator of centriole duplication during meiosis I in mouse spermatocytes, and its kinase activity is critical for this process. These observations align well with what has been reported for mitotically dividing cells (Neitzel et al, 2022). However, there are differences between spermatocytes and mitotic cells regarding PLK4 levels, localization pattern, and centriole duplication, where PLK4 remains stable and localizes to the centrosome throughout spermatogenesis, and centrioles duplicate twice during meiosis I (Fig. 1F). While pleiotropic effects of centrinone-B cannot be fully excluded, the culture conditions and drug treatment used for PLK4 inhibition did not significantly compromise cell viability and showed no effect on chromosome synapsis, centrosome integrity, or PLK4 localization in spermatocytes. Therefore, to further our understanding of PLK4 function during spermatogenesis, we next performed in vivo studies using knockout mice.

## Conditional mutation of *Plk4* causes male infertility

PLK4 is essential during embryonic development (Harris et al, 2011; Rosario et al, 2010). Therefore, to assess the effects of PLK4 depletion on centriole duplication and avoid embryonic lethality,

we used a conditional knockout (cKO) allele for *Plk4* that harbors loxP sites that flank exon 5, which encodes the kinase domain of PLK4 (LoMastro et al, 2022). To excise the floxed region of *Plk4*, we first tested the *Stra8-Cre* recombinase that conditionally knocked out *Plk4* in spermatogonia (Sadate-Ngatchou et al, 2008). However, the germ cell population was depleted prior to meiotic entry, leading to a 2.7-fold reduction in the testis to body weight ratio in adult mice. This has also been reported for cKOs of other essential kinases when using the *Stra8-Cre* transgene, including PLK1, Aurora A, and Aurora B (Wellard et al, 2020, 2021). For the remainder of the study, we utilized the *Spo11-Cre* recombinase transgene, which expresses Cre recombinase in spermatocytes at 10 days post-partum (dpp), corresponding to the preleptotene/leptotene stage of meiosis I (Lyndaker et al, 2013; Wellard et al, 2020; Hwang et al, 2018a). Mice homozygous for the *Plk4* flox allele and hemizygous for the *Spo11*-Cre transgene (*Plk4 flox/flox, Spo11-Cre tg/0*) were termed *Plk4* cKO (see Fig. 4A,B and "Methods").

Immunostaining of testis sections showed that the PLK4 signal was not detected at the centrosome in *Plk4* cKO testicular cells (Fig. 4C). There was no significant difference in the testis to body weight ratios of control and *Plk4* cKO mice ranging from 13 to 29 days post-partum (dpp) or adults (Fig. 4D,E). Furthermore, based on the assessment of cleaved caspase-3, we did not observe a significant difference in the incidence of apoptosis between control and *Plk4* cKO (Fig. EV1A–F). However, *Plk4* cKO mice were infertile (no litters were produced when four *Plk4* cKO males were each bred to two fertile WT females for a minimum of 8 weeks). Assessment of hematoxylin and eosin (H&E) stained testis cross sections revealed that *Plk4* cKO mouse testes had a significantly larger lumen to seminiferous tubule diameter ratio than the control testes (Fig. 4F,G). The larger lumen diameter was because the *Plk4* cKO spermatozoa lacked flagella (Fig. 4F). Assessment of epididymitis cross sections indicated that there were very few spermatozoa detected for the *Plk4* cKO, and those that were present appeared abnormal compared to the control (Fig. 4H). Epididymal sperm counts revealed that *Plk4* cKOs had approximately 500 times fewer spermatozoa than control mice (Fig. 4I).

To determine whether PLK4 is required for oogenesis, we excised the floxed *Plk4* allele in preleptotene oocytes, primordial follicles, and primary follicles with *Spo11-Cre*, *Gdf9-Cre*, and *Zp3-Cre*, respectively (Lan et al, 2004; Lyndaker et al, 2013; Wellard et al, 2020; Hwang et al, 2017). Unlike what was observed in males, all three *Plk4* cKO female mouse models were fertile (four *Plk4* cKO females for each Cre driver were bred to fertile WT males, and all produced three litters of 5–8 pups over a 12-week period). Genotyping of isolated oocytes demonstrated that the *Plk4* flox allele was successfully excised when Cre recombinase was expressed (Fig. EV2A–C). Nevertheless, *Plk4* cKO oocytes successfully formed bipolar spindles and showed no delay or error in their ability to progress from the germinal vesicle (GV) stage to the metaphase II (MII) arrest stage when compared to controls (Fig. EV2D). These assessments indicate that PLK4 is not essential for chromosome segregation in oocytes or female fertility.

Unlike spermatocytes, oocytes do not develop flagella, which requires centrioles during formation (Nigg and Raff, 2009). In addition, the spindle assembly and organization processes during meiotic divisions vary between spermatogenesis and oogenesis. While spermatocytes use a canonical centrosome containing two centrioles for spindle assembly and organization, oocytes lack

centrioles (Manandhar et al, 2005). Instead, oocytes rely on forming acentriolar microtubule organizing centers (aMTOCs) that consist of PCM components and liquid-like spindle domain proteins, which are critical for successful chromosome segregation following meiotic resumption (So et al, 2019). Thus, we postulate that PLK4 depletion in oocytes does not have the same effects on fertility as observed in spermatocytes due to the sexually dimorphic nature of MTOCs.

## Conditional mutation of *Plk4* in primary spermatocytes leads to centriole duplication and maturation failure

As the floxed region of the *Plk4* cKO allele was excised during the preleptotene/leptotene stage of meiosis I via *Spo11-Cre* transgene expression, we closely assessed features of meiotic prophase progression. A previous report showed that a gene-trap mutation of *Cep63*, which encodes for a pericentriolar component, resulted in homologous recombination and chromosome synapsis defects that led to germ cell apoptosis during meiotic prophase (Marjanović et al, 2015). In contrast, we observed no apparent difference in double-strand break formation, repair, or sex body formation when comparing control and *Plk4* cKO prophase I spermatocyte chromatin spreads immunolabelled for γH2AX (Fig. EV3A). Axial element formation, homologous chromosome synapsis, and desynapsis in *Plk4* cKO spermatocytes were equivalent to that of the control (Fig. EV3A). In addition, we found no difference in MLH1 foci, which mark the majority (90–95%) of crossover recombination sites, in pachytene stage spermatocytes (Fig. EV3B,C) (Anderson et al, 1999). Lack of meiotic recombination and synapsis defects (Fig. EV3A–C) or increased germ cell apoptosis (Fig. EV1) differs from what was reported for mice harboring the *Cep63* gene-trap allele (Marjanović et al, 2015). This may be explained by the fact that the *Cep63* gene-trap mutation was present within all mouse cells, whereas the *Plk4* cKO only occurred in primary spermatocytes. For instance, the *Cep63* gene-trap mutation leads to microcephaly (Marjanović et al, 2015), which could affect the regulation of the hypothalamus–pituitary gonadal axis (Cerbone et al, 2020).

We next sought to determine whether PLK4 depletion specifically affected centriole biogenesis in spermatocytes. We first assessed centriole numbers during meiotic prophase in control spermatocytes. We assessed centrioles using a CETN3 antibody or via expression of CETN2-GFP (Higginbotham et al, 2004). We confirmed that centriole duplication occurred normally in spermatocytes from mice harboring the *Cetn2-Gfp* transgene (Fig. EV3D). In agreement with previous reports (Wellard et al, 2021; Alfaro et al, 2021), control spermatocytes entered meiosis with two centrioles and underwent centriole duplication in late leptonema (Fig. 5A,B). Control spermatocytes harbored four centrioles throughout the subsequent sub-stages of meiosis I (Fig. 5A,B). While *Plk4* cKO spermatocytes entered meiosis at leptonema with two centrioles like control spermatocytes, the majority of them failed to undergo centriole duplication during late leptonema, and by early zygonema, spermatocytes continued to harbor only two centrioles (Fig. 5A,B). At metaphase I, 75.4% of spermatocytes contained only two centrioles (Fig. 5C–F). In addition, it was observed that 20.3% of *Plk4* cKO spermatocytes in metaphase I harbored only one centriole (Figs. 5C–F and EV3E,F). Centriole duplication failure during meiotic prophase

  

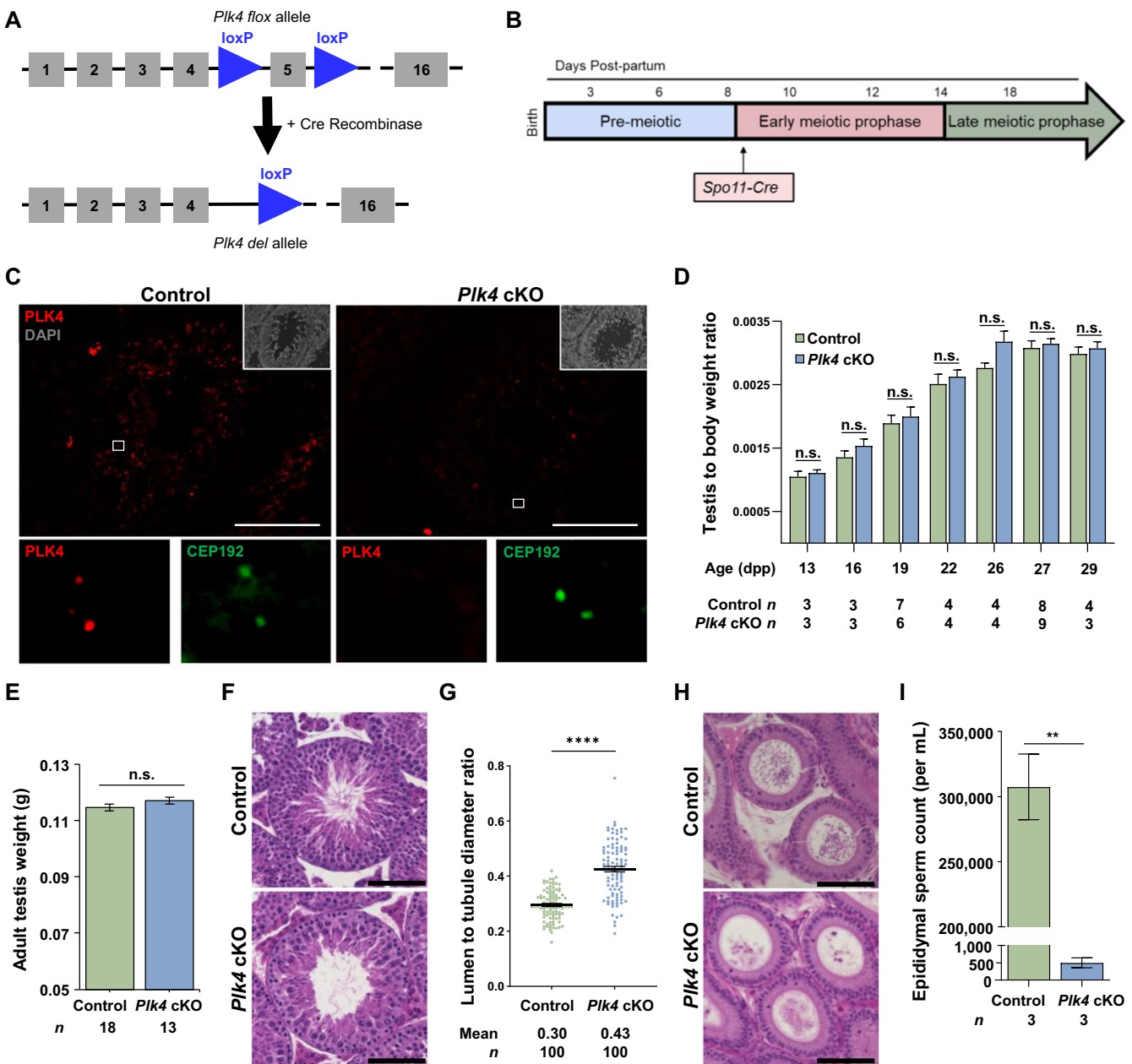

**Figure 4. Basic characterization of the testes and epididymides from *Plk4* cKO mice.**

(**A**) Diagram of the *Plk4* cKO allele before and after Cre-mediated recombination. The *Plk4* flox allele harbors two loxP sites (blue triangle) flanking exon 5 (gray box). Excision of exon 5 by Cre recombinase results in the *Plk4* deletion allele (*Plk4 del*). (**B**) The *Spo11* promoter used to drive Cre recombinase expression in the *Plk4* cKO mouse model is expressed in early meiotic prophase at ~9 dpp. (**C**) Cryosections (16 μm thick) of adult control and *Plk4* cKO (26 weeks) mouse testis immunolabeled against PLK4 (red) and stained with DAPI (gray inset in the upper right corner). Zoomed images of PLK4 and CEP192 (green) localization are shown below each corresponding image. Scale bars = 100 μm. (**D**) Quantification of the average testis to body weight ratio of control and *Plk4* cKO mice. Measurements were performed using ≥3 mice for 13, 16, 19, 22, 26, 27, and 29 dpp. *P* values for 13 dpp, 16 dpp, 19, 22, 26, and 29 dpp were 0.5928, 0.2907, 0.6027, 0.5581, 0.0817, and 0.5912, respectively. (**E**) Quantification of the average testis to body weight ratio of control and *Plk4* cKO mice. Measurements were performed using ≥3 adult mice ages 60 to 173 dpp. *P* value = 0.47. (**F**) H&E staining of 5 μm thick testis sections of 65 dpp control and *Plk4* cKO mice. Scale bar = 50 μm. (**G**) Quantification of the average lumen diameter to tubule diameter ratio in control and *Plk4* cKO mice. The average tubule diameter was 412.7 μm and 422.0 μm in control and *Plk4* cKO mice, respectively. The average tubule lumen diameter was 122.6 μm and 180.5 μm in control and *Plk4* cKO mice, respectively. Quantification was performed in three biological replicates with ≥33 tubules quantified per replicate. The total number of tubules measured for the control and *Plk4* cKO was 100 each. *P* value = <0.0001. (**H**) H&E staining of 5 μm thick epididymide sections of 95 dpp control and *Plk4* cKO mice. Scale bar = 50 μm. (**I**) Epididymal sperm count in control and *Plk4* cKO mice, respectively. The average sperm count was 307,500 sperm/mL and 500 sperm/mL in control and *Plk4* cKO epididymides, respectively. Quantification was performed using three biological replicates. *P* value = 0.0067. Data information: For all graphs (**D**, **E**, **G**, **I**), error bars show mean ± SEM. *P* values were obtained from two-tailed Student's *t* test. n.s. (not significant), **P* < 0.01, and *****P* < 0.0001.

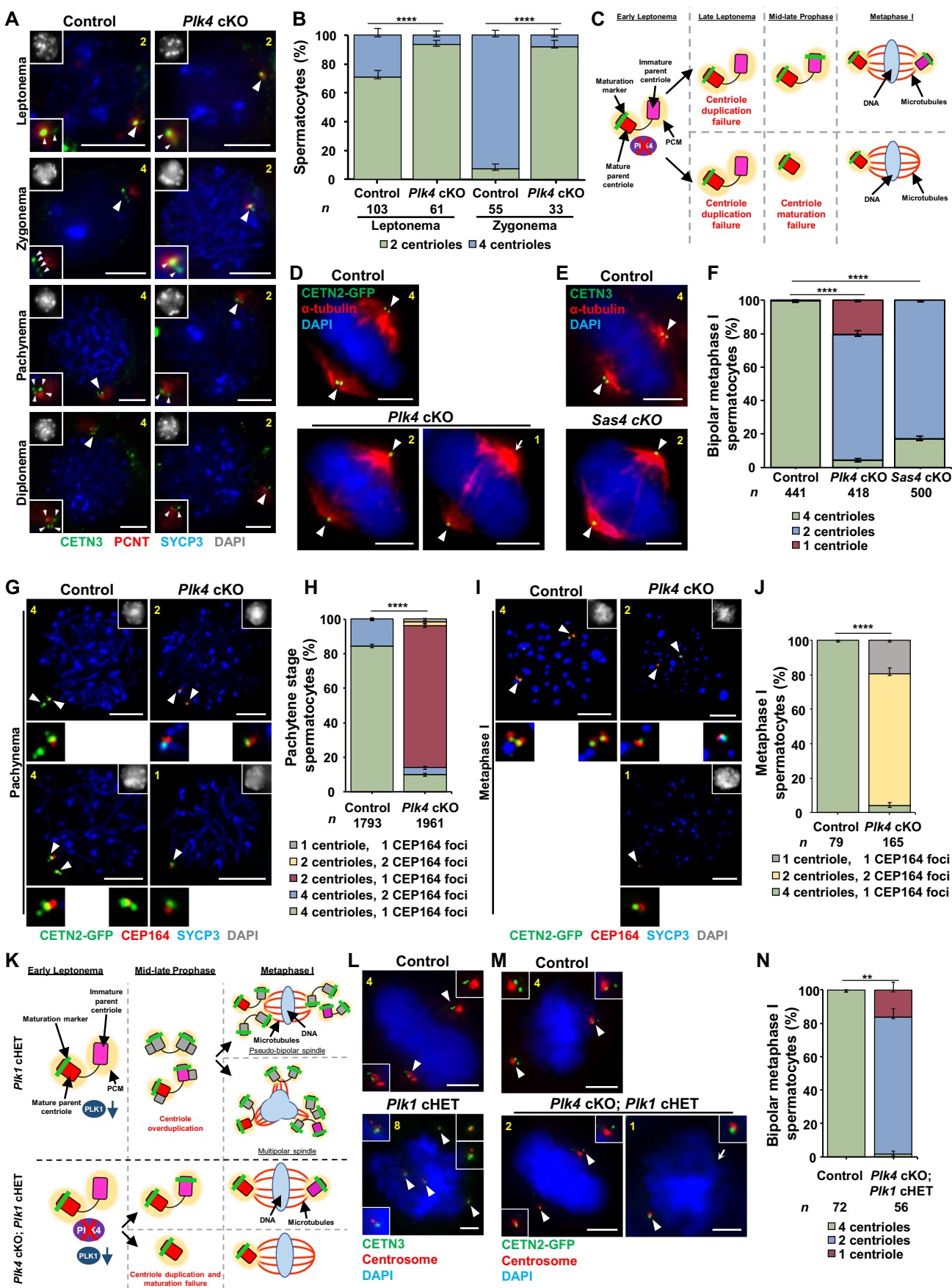

    This is a U.S. Government work and not under copyright protection in the US; foreign copyright protection may apply

**Figure 5.   Assessment of centriole duplication and maturation in *Plk4* cKO mice.**

(A) Prophase I control and *Plk4* cKO spermatocytes from 13 to 27 dpp mice were immunolabeled against CETN3 (green), PCNT (red), and SYCP3 (blue) and stained with DAPI (gray inset in upper left corner). The white arrowheads indicate the centrosome. Zoomed images of the centrioles are inset on the corresponding images. White arrowheads indicate the centrosome. The yellow number indicates the total number of centrioles per cell. Scale bars = 5 µm. (B) Quantification of centriole foci observed during leptonema and zygonema in both control and *Plk4* cKO spermatocytes. Immunolabeling was performed on three biological replicates with ≥10 spermatocytes quantified per replicate. The total number of cell quantified for control and *Plk4* cKO mice were 158 and 99, respectively. *P* values for leptonema and zygonema were XXX and XXXX, respectively. (C) Diagram illustrating how *Plk4* cKO leads to both centriole duplication and maturation failure. As a result, metaphase I spermatocytes were observed to harbor either one centriole at each bipolar spindle pole, or a single centriole at one of the two bipolar spindle poles. Red rectangle with rounded corners = mature parent centriole, pink rectangle with rounded corners = immature parent centriole, green bar = maturation marker, yellow oval = PCM, purple oval = PLK4, red X = indicates depletion, blue oval = DNA, red lines = microtubules. (D) Metaphase I control and *Plk4* cKO spermatocytes harboring CETN2-GFP (green) were immunolabeled against α-tubulin (red) and stained with DAPI (blue). The white arrowheads indicate the centrosome. The white arrow indicates the acentriolar spindle pole. The yellow number indicates the total centrioles per cell. Scale bars = 5 µm. (E) Metaphase I control and *Sas4* cKO spermatocytes were immunolabeled against CETN3 (green), α-tubulin (red) and stained with DAPI (blue). The white arrowheads indicate the centrosome. The yellow number indicates the total centrioles per cell. Scale bars = 5 µm. (F) Quantification of centrin foci per control, *Plk4* cKO, and *Sas4* cKO metaphase I spermatocytes. Immunolabeling was performed on three biological replicates, with ≥100 spermatocytes quantified per replicate. The total number of cells quantified for control, *Plk4* cKO, and *Sas4* cKO mice was 441, 418, and 500, respectively. *P* values for *Plk4* cKO and *Sas4* cKO were <0.0001 and <0.0001, respectively. (G) Pachytene stage control and *Plk4* cKO spermatocytes harboring CETN2-GFP (green) were immunolabeled against CEP164 (red), SYCP3 (blue) and stained with DAPI (gray inset in the upper right corner) Zoomed images of the centrioles are shown below the corresponding images. The white arrowheads indicate the centrosome. The yellow number indicates the total centrioles per cell. Scale bars = 5 µm. (H) Quantification and localization of CEP164 foci in relation to centrioles in control and *Plk4* cKO pachytene stage spermatocytes. Immunolabeling was performed on three biological replicates with ≥323 spermatocytes quantified per replicate. The total number of cells quantified for control, and *Plk4* cKO mice were 1793 and 1961, respectively. *P* value = <0.0001. (I) Metaphase I stage control and *Plk4* cKO spermatocytes expressing CETN2-GFP were immunolabeled against CEP164 (red) and SYCP3 (blue) and stained with DAPI (gray inset in the upper right corner). Zoomed images of the centrioles are shown below the corresponding images. The white arrowheads indicate the centrosome. The yellow number indicates the total number of centrioles per cell. Scale bars = 5 µm. (J) Quantification and localization of CEP164 foci in relation to centrioles in control and *Plk4* cKO metaphase I spermatocytes. Immunolabeling was performed on three biological replicates, with ≥12 spermatocytes quantified per replicate. The total number of cells quantified for control and *Plk4* cKO mice was 79 and 165, respectively. *P* value = <0.0001. (K) Diagram comparing centriole duplication in *Plk1* cHET and *Plk4* cKO; *Plk1* cHET spermatocytes. *Plk1* cHET spermatocytes undergo centriole overduplication, which leads to metaphase I spermatocytes with either pseudobipolar or multipolar spindle poles. In contrast, *Plk4* cKO; *Plk1* cHET spermatocytes undergo centriole duplication and maturation failure, in line with what is observed in *Plk4* cKO spermatocytes. *Plk4* cKO; *Plk1* cHET metaphase I spermatocytes harbor either one centriole at each bipolar spindle pole, or a single centriole at one of the two bipolar spindle poles. Red rectangle with rounded corners = mature parent centriole, pink rectangle with rounded corners = immature parent centriole, green bar = maturation marker, yellow oval = PCM, purple oval = PLK4, red X = indicates depletion, teal oval = PLK1, teal arrow = indicates reduction, gray rectangle with rounded corners = new centrioles from first centriole duplication, blue oval = DNA, red lines = microtubules. (L) Metaphase I control and *Plk1* cHET spermatocytes were immunolabeled against CETN3 (green) and CEP164 (red) and stained with DAPI (blue). Zoomed images of the centrioles are inset on the corresponding images. The white arrowheads indicate the centrosome. The yellow number indicates the total number of centrioles per cell. Scale bars = 5 µm. (M) Metaphase I control and *Plk4* cKO; *Plk1* cHET spermatocytes were immunolabeled against CETN3 (green) and CEP192 (red), and DNA was stained with DAPI (blue). Zoomed images of the centrioles are inset on the corresponding images. The white arrowheads indicate the centrosome. The white arrow indicates the acentriolar spindle pole. The yellow number indicates the total number of centrioles per cell. Scale bars = 5 µm. (N) Quantification of centrin foci per control and *Plk4* cKO; *Plk1* cHET metaphase I spermatocytes. Immunolabeling was performed on 3 biological replicates with ≥15 spermatocytes quantified per replicate. The total number of cells quantified for control and *Plk4* cKO; *Plk1* cHET mice was 72, and 56, respectively. *P* value = 0.002. Data information: For all graphs (B, F, H, J, N), error bars show mean ± SEM. *P* values were obtained from two-tailed Student's *t* test. **$P < 0.01$, ****$P < 0.0001$.

was not completely penetrant in the *Plk4* cKO. We hypothesized that this was due to the overlap in timing of the *Spo11-Cre* expression and the first round of centriole duplication during spermatogenesis. Nevertheless, the *Plk4* cKO centriole biogenesis defects were pronounced and significantly discernable from the control.

Based on research with mammalian cell lines, centrioles that fail to undergo maturation cannot serve as an independent MTOC, become unstable, and are disassembled (Wang et al, 2017; Karasu et al, 2022). Therefore, we hypothesized that the depletion of PLK4 caused a proportion of the centrioles to remain immature and degenerate (Fig. 5C). To further examine this hypothesis, we probed for CEP164, a distal appendage component of the centriole that associates with centrioles during maturation (Nigg and Holland, 2018). WT spermatocytes enter meiosis with two centrioles, the mature parent centriole that harbors distal and subdistal appendages and the immature parent centriole that does not contain any of these maturation markers (Wellard et al, 2021; López-Jiménez et al, 2022). At pachynema, cells with a single maturation marker colocalizing with the mature parent centriole and cells with two maturation markers colocalizing with both parent centrioles were observed (Fig. 5G,H). By metaphase I, CEP164 colocalized with all four centrioles in control

spermatocytes (Fig. 5I,J). These results correspond with our previous characterization of centriole maturation (Wellard et al, 2021; López-Jiménez et al, 2022). Analysis of *Plk4* cKO spermatocytes at pachynema and metaphase I indicated that a population of spermatocytes had delayed and defective centriole maturation, which led to the degeneration of the immature parent centriole (Fig. 5G–J).

We next assessed whether the centriole maturation defect and subsequent centriole loss were due to the inability to build new centrioles or specifically due to the loss of PLK4. To address this, we utilized male mice that harbored a *Sas4* (*CenpJ*) cKO allele (*Sas4* flox/flox, *Spo11*-Cre tg/0; termed *Sas4* cKO) (Bazzi and Anderson, 2014). SAS4 is a structural component of the centriole and is required during the assembly of new centrioles (McIntyre et al, 2012; Leidel and Gönczy, 2003). In *Sas4* cKO mice, 82.8% of spermatocytes failed to duplicate centrioles during meiotic prophase, leading to the formation of spermatocytes containing only two centrioles (Fig. 5E,F). However, we never observed *Sas4* cKO spermatocytes that only had a single centriole. This suggests that centrioles still undergo centriole maturation despite centriole duplication failure. Taken together, these results indicate that PLK4 is directly responsible for centriole duplication and maturation processes during spermatogenesis.

## PLK4 functions upstream of PLK1 during centriole duplication

We had previously observed that another PLK family member, PLK1, was essential for centrosome biogenesis (Wellard et al, 2021). Conditional heterozygous (cHet) mutation of *Plk1* resulted in more than one round of centriole duplication during meiotic prophase (Fig. 5K,L). This indicates that PLK1 is also important for regulating centriole duplication during spermatogenesis. To determine whether the centriole overduplication phenotype in *Plk1* cHet spermatocytes is dependent on PLK4 function, we created mutant mice that harbored compound mutations of *Plk1* cHet and *Plk4* cKO (*Plk4* flox/flox, *Plk1* +/flox, *Spo11*-cre tg/0; termed *Plk4* cKO; *Plk1* cHET). We assessed meiosis I spermatocytes and observed that centriole duplication was blocked in *Plk4* cKO; *Plk1* cHET mice (Fig. 5K–N). This demonstrated that PLK4 function is required to stimulate centriole amplification when PLK1 levels are reduced. Thus, PLK4 functions upstream of PLK1, and PLK4 is the master regulator of centriole biogenesis during mammalian spermatogenesis.

## Non-centriolar spindle poles form in *Plk4* cKO spermatocytes during meiosis I

Despite *Plk4* cKO spermatocytes failure to duplicate and mature centrioles during meiotic prophase, they still progressed to form a bipolar spindle at metaphase I (Fig. 5D,F). This was intriguing because centrioles are a core centrosome component, and the centrosome is the primary MTOC in spermatocytes (Chemes, 2012). However, there are other cases where the centrioles are not required for bipolar spindle pole formation in mammalian cells (Meunier and Vernos, 2016; So et al, 2019; Coquand et al, 2021). For instance, oocytes rely on aMTOCs, which comprise PCM components that coalesce to organize the metaphase I and II bipolar spindles (So et al, 2019). Based on this alternative method of spindle organization, we hypothesized that spermatocytes may use a similar aMTOC mechanism as a backup to ensure that chromosome segregation successfully takes place. Therefore, we assessed the localization of PCM proteins that are known to be components of aMTOCs in oocytes.

Assessment of PCM components CEP192, CDK5RAP2, PCNT, NEDD1, and components of the γ-TURC complex (γ-tubulin, GCP2, and GCP4), in metaphase I control spermatocytes demonstrated they all colocalized with the centrioles at both spindle poles (Figs. 6A and EV4A–E). However, in *Plk4* cKO spermatocytes containing a single centriole, PCM components colocalized with the centriole but were not present at the acentriolar spindle pole (Figs. 6A–C and EV4A–E). Among these observations, the most striking to us was the absence of the γ-TURC components, which is canonically required for microtubule nucleation (Roostalu and Surrey, 2017; Sulimenko et al, 2022). The absence of γ-TURC and PCM components indicated that the spindle pole was not only acentriolar but also lacked major centrosomal components. These observations demonstrate that spermatocytes are capable of forming a non-centrosomal spindle pole when centriole duplication and maturation fail.

To further characterize aspects of the non-centrosomal spindle pole, we compared the length and width of spindle poles in both control and *Plk4* cKO metaphase I spermatocytes (Fig. 6B and

"Methods"). The spindle pole on the non-centrosomal side of *Plk4* cKO metaphase I spermatocytes was reduced in both length and width compared to the canonical spindle pole in metaphase I spermatocytes (Fig. 6B,D). We also observed a 2.2-fold increase in the number of chromosome missegregation events in *Plk4* cKO anaphase I spermatocytes compared to controls (Fig. EV3J,K). Nevertheless, *Plk4* cKO spermatocytes progressed to metaphase II harboring 2, 1, or 0 centrioles and formed spermatids without a significant increase in apoptosis (Figs. 4F–H, EV1 and EV3L), indicating that non-centrosomal MTOCs (ncMTOCs) are proficient in mediating chromosome segregation.

## ncMTOCs in spermatocytes consist of a subset of microtubule-stabilizing and organizing proteins

Our next aim was to determine what factors contribute to the formation of the non-centrosomal spindle pole in the *Plk4* cKO spermatocytes. We identified proteins that were good candidates for aiding in microtubule nucleation, organization, and stabilization and evaluated their localization in control and *Plk4* cKO metaphase I spermatocytes (Fig. 7A). One class of proteins assessed were microtubule motors, including dynein and the kinesins KIF2A, KIF18a, and KIFC1 (Mountain et al, 1999; Norris et al, 2018; Roostalu et al, 2018; Ganem and Compton, 2004; Ferenz et al, 2009; Weaver et al, 2011; van der Voet et al, 2009). We also examined ch-TOG and HURP, both of which are known microtubule stabilizers in mitotic cells, as well as microtubule severing proteins KATNB1, KATNA1, and KATNAL1 (Dudka et al, 2019; De Luca et al, 2008; Booth et al, 2011; Dunleavy et al, 2021). However, none of these molecular motors, microtubule stabilizers, and microtubule severing enzymes localized to the non-centrosomal spindle pole upon assessment via IF microscopy (Fig. 7A; Appendix Fig. S1A–W).

From our extensive analyses, however, we did identify six proteins that localized to the ncMTOC in spermatocytes: TPX2 (TPX2 microtubule nucleation factor), CAMSAP1-3 (calmodulin regulated spectrin associated proteins), KIF11 (Eg5; Kinesin Family Member 11), and NuMA (nuclear mitotic apparatus protein1) (Figs. 7B and EV4F–K). All three CAMSAP proteins are minus-end microtubule stabilizers that prevent minus-end microtubule catastrophe and promote microtubule nucleation (Mao et al, 2019; Imasaki et al, 2022; Jiang et al, 2014; Khalaf-Nazzal et al, 2022; Coquand et al, 2021). TPX2 promotes microtubule nucleation and stabilizes along the lengths of the microtubules by binding within the grooves of the alpha and beta-microtubule subunits to help prevent unwanted microtubule catastrophe (Roostalu et al, 2015; Zhang et al, 2017a). KIF11 is a plus-end-directed microtubule motor protein that is known to aid in bipolar spindle orientation (She et al, 2020, 2022; Kapitein et al, 2005). NuMA plays an important role in the formation and organization of the metaphase spindle by helping to orient microtubule minus ends to a unified location and facilitating connections between the spindle pole and the cellular membrane to ensure proper spindle orientation (Gallini et al, 2016; van der Voet et al, 2009).

TPX2 is known to be a binding partner of Aurora A kinase (AURKA), which has been implicated in spindle assembly, γ-TuRC activation, and centrosome separation (Zhang et al, 2017b; Wellard et al, 2020). We assessed multiple antibodies against AURKA to determine its localization in *Plk4* cKO spermatocytes. From these

    

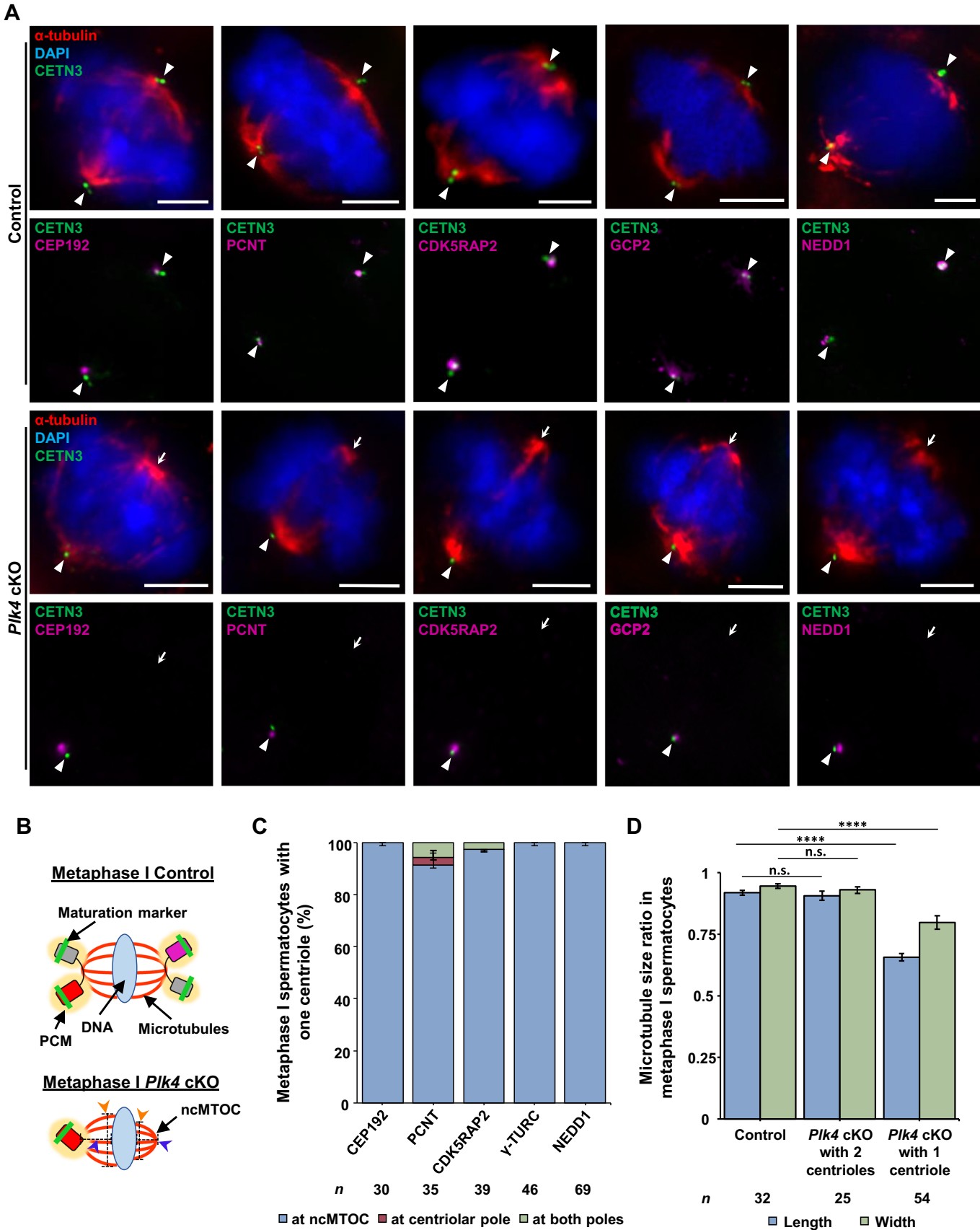

**Figure (A)** Control / *Plk4* cKO panels stained for α-tubulin, DAPI, CETN3, CEP192, PCNT, CDK5RAP2, GCP2, NEDD1.

**B** Metaphase I Control — Maturation marker, PCM, DNA, Microtubules. Metaphase I *Plk4* cKO — ncMTOC.

**C** Metaphase I spermatocytes with one centriole (%): CEP192, PCNT, CDK5RAP2, γ-TURC, NEDD1. *n* = 30, 35, 39, 46, 69. Legend: at ncMTOC, at centriolar pole, at both poles.

**D** Microtubule size ratio in metaphase I spermatocytes: Control, *Plk4* cKO with 2 centrioles, *Plk4* cKO with 1 centriole. *n* = 32, 25, 54. n.s., n.s., ****, ****. Legend: Length, Width.

◄ **Figure 6. PCM components only localize to the spindle pole containing centrioles.**

(A) Metaphase I control and *Plk4* cKO spermatocytes were immunolabeled against CETN3 (green), α-tubulin (red), CEP192, PCNT, CDK5RAP2, GCP2, and NEDD1 (purple) and stained with DAPI (blue). The white arrowheads indicate the centrosome. The white arrows indicate the ncMTOC. Scale bars = 5 μm. (B) Diagram illustrating the ncMTOC and its reduced spindle width and length in *Plk4* cKO metaphase I spermatocytes. Red rectangle with rounded corners = mature parent centriole, pink rectangle with rounded corners = immature parent centriole, green bar = maturation marker, yellow oval = PCM, gray rectangle with rounded corners = new centrioles from first centriole duplication, blue oval = DNA, red lines = microtubules, orange arrowheads and black dashed line = indicate spindle width measurement, blue arrowheads and black dashed line = indicate spindle length measurement. (C) Quantification of PCM component foci localization in metaphase I spermatocytes with only one centriole in *Plk4* cKO mice (24–27 dpp). Immunolabeling was performed on ≥3 biological replicates with ≥10 spermatocytes quantified per replicate. The total number of cells quantified and immunolabelled for CEP192, PCNT, CDK5RAP2, γ-TURC, and NEDD1 were 30, 35, 39, 46, and 69, respectively. (D) Quantification of spindle length and width ratios in control and *Plk4* cKO metaphase I spermatocytes. Measurement of spindle length and width were performed in metaphase I spermatocytes immunolabeled against α-tubulin on ≥3 biological replicates with ≥8 spermatocytes quantified per replicate. The total number of cells quantified for control spermatocytes, *Plk4* cKO spermatocytes with 2 centrosomes, and *Plk4* cKO spermatocytes with one centrosome were 32, 25, and 54, respectively. *P* values for *Plk4* cKO with 2 centrioles spindle length and width were 0.5453 and 0.4161, respectively. *P* values for *Plk4* cKO with 1 centriole spindle length and width were <0.0001 and <0.0001, respectively. Data information: For all graphs (C, D), error bars show mean ± SEM. For graph (D), *P* values were obtained from a two-tailed Student's *t* test. n.s. (not significant), and ****$P < 0.0001$.

assessments, we concluded that AURKA does not localize to the ncMTOC (Appendix Fig. S1M). However, to further assess the role of AURKA in *Plk4* cKO spermatocytes, we treated seminiferous tubules with an AURKA inhibitor, MLN8237 (see "Methods"). MLN8237 treatment did not disrupt TPX2 localization or bipolar spindle assembly in spermatocytes (Fig. 8A). *Plk4* cKO spermatocytes with a single centriole successfully formed bipolar spindles and TPX2 localized to both the centrosome and ncMTOC (Fig. 8A). As a control, we assessed the localization of another known AURKA binding partner and substrate, TACC3. Inhibition of AURKA led to the disruption of TACC3 localization to the spindle poles in mouse spermatocytes at metaphase I, as previously reported (Fig. 8A–C) (Simerly et al, 2024). This indicated that AURKA inhibition was successful, and AURKA kinase activity is not required for TPX2 localization and ncMTOC formation in *Plk4* cKO spermatocytes.

We also performed additional assessments examining the effects of inhibiting microtubule motor proteins Dynein and KIF11. While we did not observe the minus-end-directed motor protein dynein at the ncMTOC through IF studies (Appendix Fig. S1A–F), it has been implicated in NuMA localization (He et al, 2023; Merdes et al, 1996, 2000). In spermatocytes, the meiosis I specific DYNLRB2-containing dynein complex is responsible for interacting with NuMA and successfully delivering it to the spindle poles (He et al, 2023). When we treated control and *Plk4* cKO spermatocytes with the dynein inhibitor, Dynarrestin, we observed an increase in instances of disordered chromatin organization at the metaphase plate (Fig. 8D). However, NuMA localization to centrosomes or ncMTOCs was not perturbed (Fig. 8E,F). Dynein localization indicated that the inhibitor treatment was successful, as the protein no longer colocalized with the spindle (Fig. 8F,G). While dynein is not observed at the ncMTOCs in *Plk4* cKO spermatocytes, its motor function is likely important for microtubule movements contributing to spindle assembly and metaphase plate formation in spermatocytes.

Treatment of spermatocytes with the KIF11 inhibitor, ARRY-520, revealed that KIF11 is critical for maintaining a bipolar spindle during spermatogenesis. We observed control and *Plk4* cKO spermatocytes after one and two hours of ARRY-520 treatment to assess bipolar spindle assembly. After one hour, there was an increase in *Plk4* cKO spermatocytes with ncMTOCs that failed to fully form bipolar spindles (Fig. 8H). After two hours, all *Plk4* cKO spermatocytes harboring a single centriole were monopolar

(Fig. 8I,J). There was also an increase in monopolar spindles in control metaphase I spermatocytes (Fig. 8J,K). Moreover, KIF11 no longer colocalized with the spindle poles, indicating ARRY-520 treatment was successful (Fig. 8I). Taken together, this suggests that KIF11 is critical for both centrosomal and non-centrosomal bipolar spindle formation in mammalian spermatocytes.

Based on these findings, we propose that CAMSAP1-3 and TPX2 serve as the base for microtubule nucleation, promoting polymerization in the absence of γ-tubulin while helping stabilize the microtubule minus ends and preventing catastrophe at the base of the microtubules (Fig. 8L). TPX2 further contributes to microtubule stabilization along the lengths of the microtubules. KIF11 is critical for MTOC separation and the formation of a bipolar spindle. NuMA organizes the microtubules by focusing the minus ends at a unified pole and orienting that pole with the cellular membrane, ensuring the spindle is in a bipolar arrangement. Based on their localization to the centrosome and ncMTOC in spermatocytes, CAMSAP1-3, TPX2, NuMA, and KIF11 are likely key factors in MTOC regulation during spermatogenesis. Their importance is made more apparent when centriole duplication fails.

## *Plk4* cKO spermatocytes fail to successfully complete spermiogenesis

Spermiogenesis is the process by which round spermatids undergo the necessary restructuring to become spermatozoa. In mice, this includes nuclear remodeling involving chromosome compaction and nuclear polarization to one side of the cell in addition to drastic changes in the cell structure, such as the formation of the flagella and the hooked sperm head, a process in part mediated by the manchette (O'Donnell, 2014). The manchette is a microtubule-based structure that forms between the perinuclear ring at the base of the acrosome and the axoneme (Lehti and Sironen, 2016). The manchette facilitates major cell morphological changes during spermiogenesis by serving as a system for protein trafficking (Lehti and Sironen, 2016). Manchette microtubules are anchored to the perinuclear ring complex and it is thought that microtubules nucleate from this location (Lehti and Sironen, 2016; Yoshida et al, 1994).

Despite having a non-centrosomal spindle pole, *Plk4* cKO spermatocytes still progress through spermatogenesis without evidence of increased apoptosis (Fig. EV1). We observed that a majority of *Plk4* cKO round and elongating spermatids harbored

 

no centrioles (61.6% and 72.7%, respectively) compared to control spermatids that canonically have two centrioles (Fig. 9A–D). Our assessments of PLK4 localization in elongating spermatids indicated that PLK4 localized to the perinuclear ring and manchette

structures (Fig. 1H). To assess the integrity of the perinuclear ring in *Plk4* cKO spermatocytes, we probed for δ-tubulin, a known component of the perinuclear ring structure (Fig. 9E–H) (Kato et al, 2004). δ-tubulin also localized to the flagella of spermatozoa

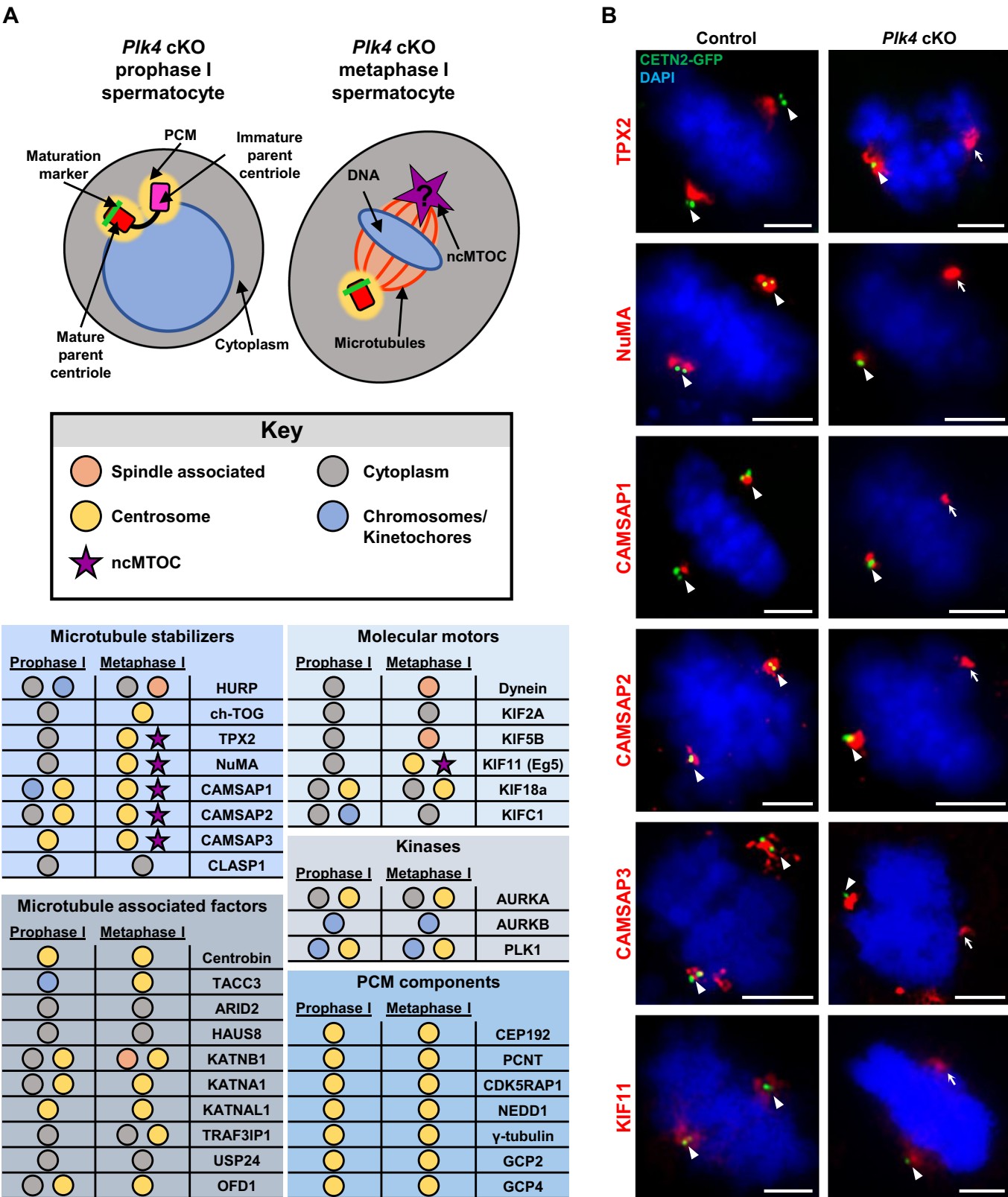

◄

Figure 7.  Identification of microtubule-associated proteins that localized to the ncMTOC.

(A) Diagram illustrating a *Plk4* cKO prophase I and metaphase I spermatocyte. Red rectangle with rounded corners = mature parent centriole, pink rectangle with rounded corners = immature parent centriole, green bar = maturation marker, yellow oval = PCM, purple star = ncMTOC, blue circle/oval = DNA, red lines = microtubules, orange shading between red lines = spindle associated region, gray circle/oval = cytoplasm. The localization of microtubule-associated proteins was assessed and reported in the table presented below the diagram key. Observed localization corresponds to the diagram and is outlined in the key as follows: spindle associated (orange circle), cytoplasm (gray circle), centrosome (yellow circle), chromosomes/kinetochores (blue circle), and ncMTOC (purple star). See Fig. EV4 for representative images of prophase I and metaphase I control and *Plk4* cKO spermatocytes immunostained for all microtubule-associated proteins listed in the table. (B) Metaphase I spermatocytes expressing CETN2-GFP were immunolabeled against TPX2, NuMA, CAMSAP1, CAMSAP2, CAMSAP3, and KIF11 (red) and stained with DAPI (blue). The white arrowheads indicate the centrosome. The white arrows indicate the ncMTOC. Scale bars = 5 μm.

(Fig. 9E). We also analyzed the coiled-coil domain-containing protein 13, CCDC13. This centriolar satellite protein has been identified in mitotically dividing cells as a critical component during ciliogenesis (Staples et al, 2014), but has not been previously assessed in meiotic cells. CCDC13 localized to the perinuclear ring and the centrin signal in control elongating spermatids (Fig. 9H). However, in *Plk4* cKO spermatocytes, δ-tubulin-positive flagella were absent, and there were defects in the perinuclear ring structure and its connection to the base of the acrosome (Fig. 9E–H). Based on α-tubulin localization, we observed no change in manchette morphology in *Plk4* cKO elongating spermatids (Fig. 9B,I). However, the manchette was remained present in *Plk4* cKO spermatozoa when it was disassembled in control spermatozoa (Fig. 9B,I). Together, the PLK4 localization pattern and *Plk4* cKO defects observed during spermiogenesis indicated that PLK4 has multiple functions during spermatid morphogenesis that are critical for the formation of elongated spermatozoa harboring functional flagella.

Control spermatids form flagella and hooked sperm heads during spermiogenesis to become spermatozoa, whereas *Plk4* cKO spermatids failed to complete this structural remodeling (Fig. 9I). Instead, *Plk4* cKO spermatozoa exhibited misshapen elongated configurations, never successfully forming the hooked head structure or flagella, including *Plk4* cKO spermatids that inherited a single centriole (Fig. 9I). Similar spermiogenesis defects were observed for the *Sas4* cKO mice (Fig. 9J). Overall, these observations indicate that PLK4 and centriole inheritance during meiosis are critical for mediating spermiogenesis. Furthermore, the defects during spermiogenesis are likely the primary cause of reduced epididymal sperm counts and infertility observed for the *Plk4* cKO male mice.

## Discussion

### Spermatocytes can utilize a non-centrosomal spindle pole to mediate chromosome segregation during meiosis

We determined that mouse spermatocytes are capable of mediating bipolar chromosome segregation in the absence of centriole and centrosome components. We discovered that CAMSAP1-3, TPX2, NuMA, and KIF11 localize to ncMTOCs and are likely key components that ensure microtubule nucleation and bipolar spindle formation (Tsuchiya and Goshima, 2021). TPX2 and CAMSAP1-3 are microtubule-stabilizing proteins that have been observed to promote microtubule nucleation in the absence of γ-tubulin (Tsuchiya and Goshima, 2021; Meunier and Vernos, 2016).

CAMSAP1-3 are microtubule minus-end stabilizers that bind with microtubules to prevent catastrophe at the sites closest to microtubule nucleation (Mao et al, 2019; Jiang et al, 2014). CAMSAP1 and 2 have also been shown to serve as nucleation sites for microtubules in cells undergoing mitotic division (Coquand et al, 2021; Imasaki et al, 2022). Additionally, in glial cells, microtubules were observed to nucleate from foci of GFP-labeled CAMSAP1 and 2 within the basal process region where γ-tubulin was undetectable (Coquand et al, 2021). TPX2 is also a microtubule stabilizer that binds more evenly along the lengths of microtubules to help prevent microtubule catastrophe, in addition to helping promote microtubule nucleation in the absence of γ-tubulin (King and Petry, 2020; Schatz et al, 2003; Zhang et al, 2017a; Roostalu et al, 2015). NuMA is responsible for helping orient the spindle in relation to the cell membrane and aids in the focusing of microtubule minus ends to a unified point (Gallini et al, 2016; van der Voet et al, 2009). Lastly, KIF11 has been shown to be required for centrosome separation and bipolar spindle orientation in mitotic and meiotically dividing cells (She et al, 2020). This, plus-end-directed microtubule motor protein, provides an outward pushing force on MTOCs through the cross-linking of antiparallel microtubules (She et al, 2022; Raaijmakers et al, 2012). Together, these six factors play key roles in successful spindle pole formation and function (Fig. 8L). Future assessments will be conducted to determine how these, and additional factors yet to be identified, coordinate ncMTOC formation during spermatogenesis.

### *Plk4* cKO mice are infertile due to defects during spermiogenesis

Despite being able to successfully complete meiosis I and II in the absence of centriole duplication, *Plk4* cKO mice were still infertile. We attribute this phenotype to a failure during spermiogenesis. During spermiogenesis, spermatids must undergo major structural reconfiguration, forming both a head and tail structure needed for spermatozoa function during fertilization. Similar to other ciliary structures, the centriole acts as the base structure for the microtubule axoneme to polymerize and extend to form the flagella (Avidor-Reiss et al, 2019, 2020). In mouse and human spermatocytes, the two centrioles are oriented perpendicular to one another in order to successfully anchor to what will become the head of the spermatozoa and serve as the base structure for the axoneme to extend from (Chemes, 2012; Manandhar et al, 1998). Both *Plk4* and *Sas4* cKO result in the formation of spermatids that do not harbor a centriole pair. Therefore, it was expected that *Plk4* and *Sas4* cKO spermatocytes undergoing spermiogenesis failed to form flagella. *Plk4* cKO spermatids also exhibited defects in sperm head

   

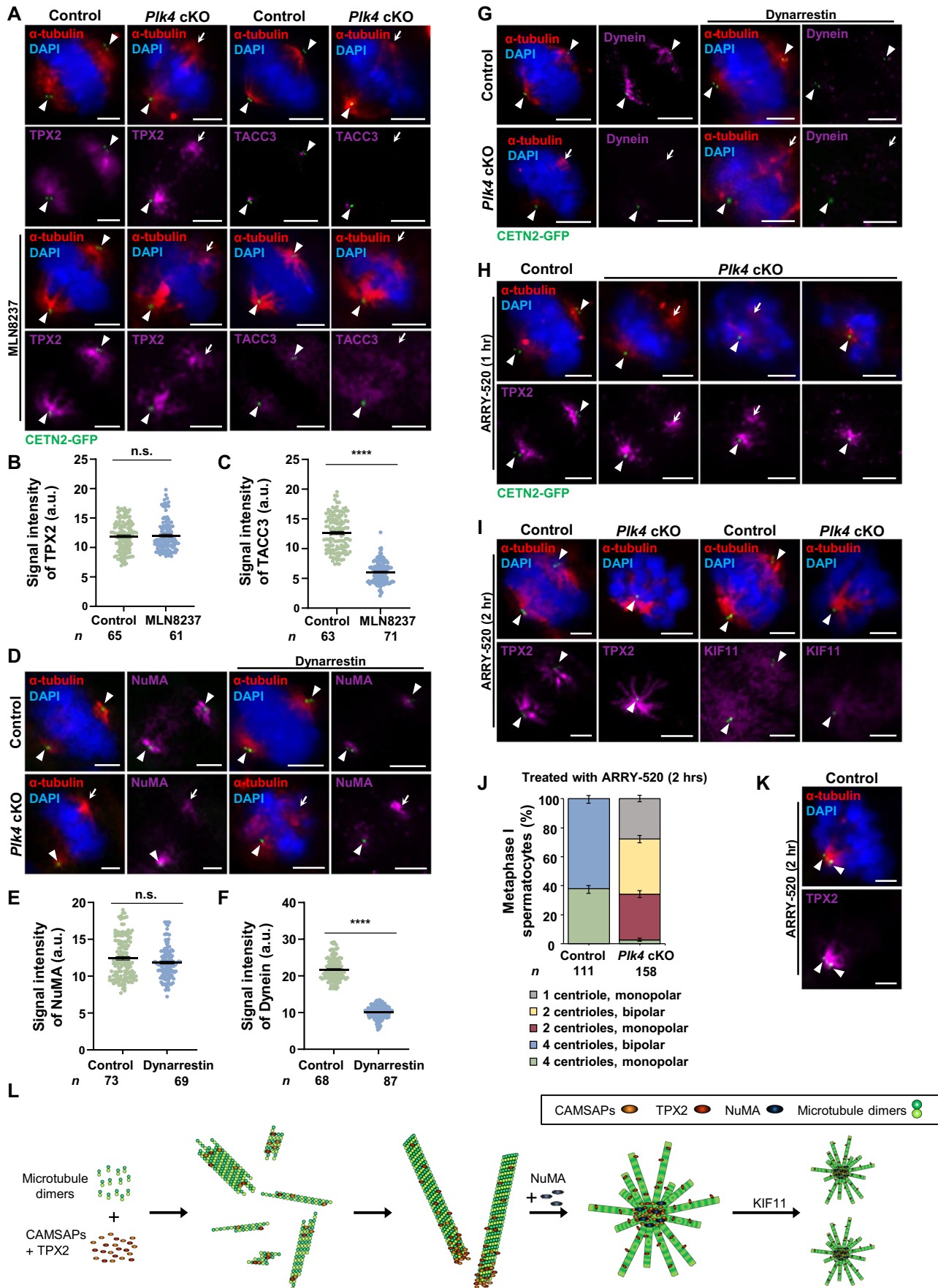

morphogenesis. We found that PLK4 localizes to the pericentriolar ring and the manchette structures (Fig. 1H). Due to this observed localization beyond the centrosome, PLK4 is likely involved in additional aspects of cellular restructuring during spermiogenesis beyond flagella formation. Although centrioles are disassembled in rodents during spermiogenesis, we show that centrioles are required for flagella formation and successful execution of spermiogenesis, just as they are in human spermatozoa (Manandhar et al, 1998, 1999).

## Centriole biogenesis in non-mammalian organisms

PLK4 localization to the centrosome throughout mammalian spermatogenesis suggests that the kinase is regulated differently in spermatocytes compared to mitotic cells. A difference in PLK4 regulation between mitotic and meiotic cell types is further supported by research on the *Caenorhabditis elegans* ortholog of PLK4, ZYG-1. A truncated version of ZYG-1 that was no longer capable of localizing to mitotic centrosomes was still able to localize to the centrosome in spermatocytes (Peters et al, 2010). This suggests that PLK4 localization to the centrosome is mediated differently in meiotic vs mitotic cell types. Identifying spermatocyte-specific interaction partners of PLK4 may shed light on how it remains localized to the centrosome throughout both meiotic divisions.

It has been previously reported that the *Drosophila melanogaster* PLK4 ortholog, SAK, is required for centriole duplication in premeiotic and meiotic cells during spermatogenesis (Bettencourt-Dias et al, 2005). When SAK was depleted via RNAi, centriole duplication failed but cells progressed through spermatogenesis (Bettencourt-Dias et al, 2005). It was shown that centrosomal components, D-PLP (PCNT) and γ-tubulin, were absent from spindle poles that lacked centrioles (Bettencourt-Dias et al, 2005).

Also, in line with our observations, SAK-depleted spermatocytes were unable to successfully complete spermiogenesis, resulting in sperm lacking flagella (Bettencourt-Dias et al, 2005). Despite the similarities between our work and the reported findings in *D. melanogaster*, there are some key differences. First, depletion of SAK resulted in centriole duplication failure in the germ line, prior to entering meiosis (Bettencourt-Dias et al, 2005). They observed meiosis I spermatocytes that were devoid of centrioles, indicating duplication failure had taken place prior to meiotic entry (Bettencourt-Dias et al, 2005). Therefore, the role of SAK in relation to centriole duplication was not assessed in strictly a meiotic context. Secondly, centriole duplication only occurs once during *D. melanogaster* spermatogenesis at the beginning of the first prophase, whereas centrioles duplicate twice in mouse and human spermatocytes (Riparbelli et al, 2018; Bettencourt-Dias et al, 2005; Wellard et al, 2021; Breslow and Holland, 2019). *D. melanogaster* spermatids canonically complete spermiogenesis after inheriting a single centriole. Mammalian spermatids each inherit two centrioles. Our results demonstrate that both centrioles are required for successful spermiogenesis, as *Plk4* and *Sas4* cKO spermatids that harbor less than two centrioles failed to form flagella (Fig. 9I). Lastly, the expression patterns of PLK4 during spermatogenesis also vary between mice and *D. melanogaster*. While we observed PLK4 at the centrosome throughout spermatogenesis in mice (Fig. 1H), SAK levels are reduced as spermatogenesis progresses in *D. melanogaster* (Khire et al, 2015). This reduction in SAK is a requirement for successful *D. melanogaster* embryo development after fertilization (Khire et al, 2015; Varmark et al, 2007; Sullenberger et al, 2023; Hatch et al, 2010) The differences in centriole duplication processes, PLK4/SAK expression levels, and PLK4/SAK localization patterns between *D. melanogaster* and mouse spermatogenesis highlight the importance of assessing the roles of PLK4 in mammalian spermatocytes.

 

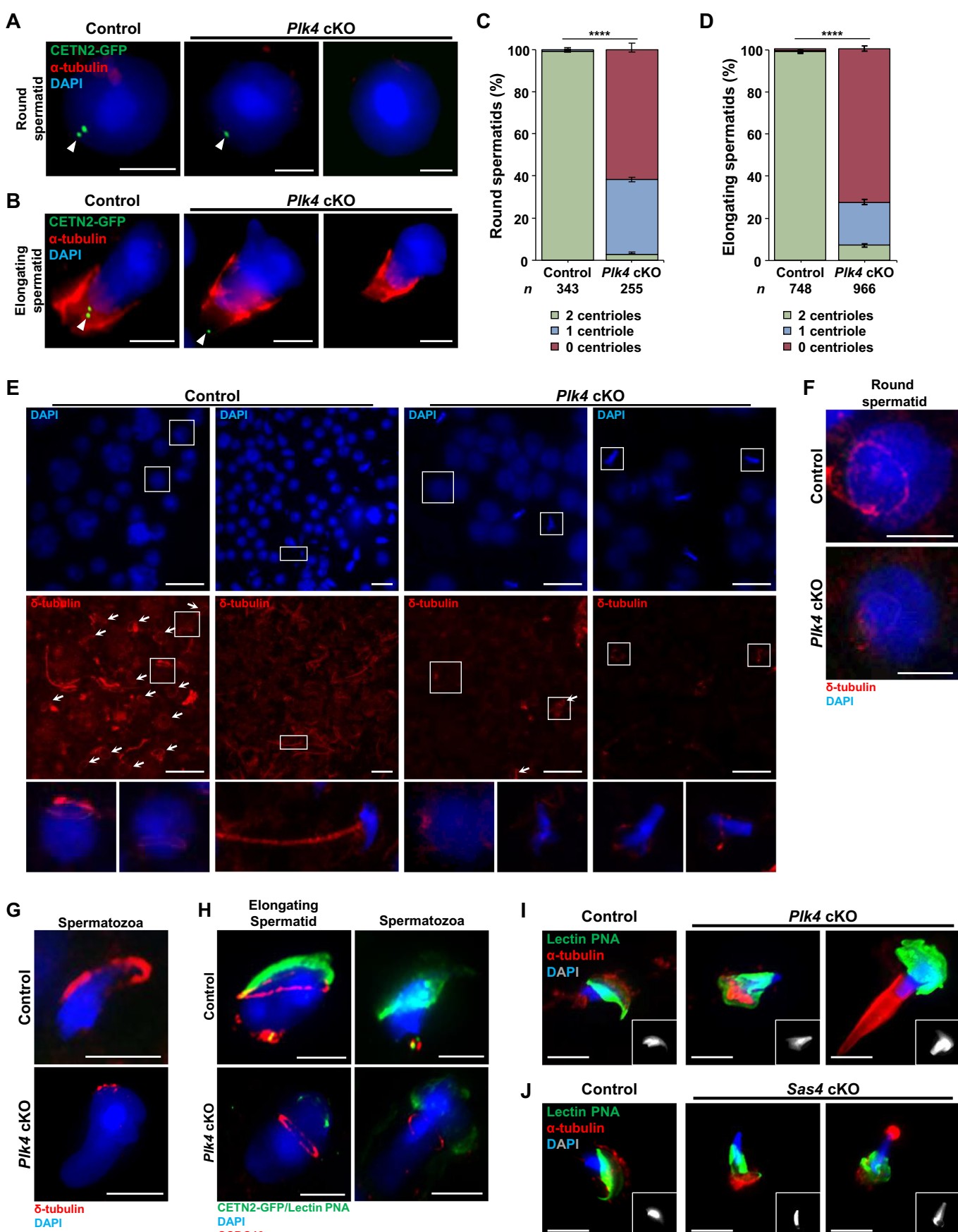

**Figure 9. Assessment of spermiogenesis in *Plk4* cKO mice.**

(A) Control and *Plk4* cKO round spermatids expressing CETN2-GFP (green) were immunolabeled against α-tubulin (red) and stained with DAPI (blue). The white arrowheads indicate the centrosome. Scale bars = 5 μm. (B) Control and *Plk4* cKO elongating spermatids expressing CETN2-GFP (green) were immunolabeled against α-tubulin (red) and stained with DAPI (blue). The white arrowheads indicate the centrosome. Scale bars = 5 μm. (C) Quantification of centrin foci in control and *Plk4* cKO round spermatids. Immunolabeling was performed on three biological replicates with ≥20 spermatocytes quantified per replicate. The total number of cells quantified for control, and *Plk4* cKO mice were 343 and 255, respectively. *P* value = <0.0001. (D) Quantification of centrin foci in control and *Plk4* cKO elongating spermatids. Immunolabeling was performed on three biological replicates, with ≥36 spermatocytes quantified per replicate. The total number of cells quantified for control and *Plk4* cKO mice was 748 and 966, respectively. *P* value = <0.0001. (E) Representative image of adult control and *Plk4* cKO spermatocytes from mice immunolabeled against δ-tubulin (red) and stained with DAPI (blue). The white arrows indicate perinuclear ring structure. Zoomed image of spermatozoa and tail inset on the corresponding image. Scale bars = 25 μm. (F) Control and *Plk4* cKO round spermatids from mice immunolabeled against δ-tubulin (red) and stained with DAPI (blue). Scale bars = 5 μm. (G) Control and *Plk4* cKO spermatozoa from mice immunolabeled against δ-tubulin (red) and stained with DAPI (blue). Scale bars = 5 μm. (H) Elongating spermatid and spermatozoa control and *Plk4* cKO spermatocytes harboring CETN2-GFP (green) were immunolabeled against lectin–PNA (green), CCDC13 (red), and stained with DAPI (blue). Scale bars = 5 μm. (I) Control and *Plk4* cKO spermatozoa were immunolabeled against lectin–PNA (green), α-tubulin (red), and stained with DAPI (blue, gray inset in lower right corner). Scale bars = 5 μm. (J) Control and *Sas4* cKO spermatozoa were immunolabeled against lectin–PNA (green), α-tubulin (red), and stained with DAPI (blue, gray inset in lower right corner). Scale bars = 5 μm. Data information: For all graphs (C, D), error bars show mean ± SEM. *P* values were obtained from two-tailed Student's *t* test. ****P* < 0.0001.

## Centrioles and human disease

Previously, the role of mammalian PLK4 had only been assessed in the context of mitotic division. Our work demonstrated that PLK4 is critical for successful centriole duplication and maturation during spermatogenesis and was also essential for normal sperm morphogenesis. We showed similar PLK4 localization patterns in mouse and human spermatocytes, which suggests that the function of PLK4 is conserved between species. Importantly, PLK4 has been linked with cases of human infertility. It has been reported that individuals with a 13 base pair deletion or a missense mutation in *PLK4* exhibit a combination of non-obstructive azoospermia and Sertoli cell only (SCO) syndrome (Miyamoto et al, 2016; Tang et al, 2022; Muranishi et al, 2024). Our characterization of PLK4 in mice can contribute to developing better diagnostic and treatment plans for patients with PLK4 and centrosome-related fertility defects.

Conditions resulting from centriole aberrancies, or ciliopathies, are most commonly associated with respiratory system deficiencies or other organs in the body that rely heavily on multiciliate cell types (Inaba and Mizuno, 2016). However, ciliopathies can manifest in the context of flagella formation and, thus, can result in fertility issues (Linck et al, 2016). Conditions such Bardet-Biedl and Kartagener syndromes are examples of such conditions where subfertility is an observed phenotype in patients (Lindstrand et al, 2016; Sha et al, 2014; Halbert et al, 1997). Patients with these conditions experience primary ciliary dyskinesia or the abnormal/absent cilia motion that has detrimental impacts on not only the flagella of spermatozoa in males, but also on the ciliated cells along the female reproductive tract (Newman et al, 2023). Characterization of PLK4 in the context of spermatogenesis, as well as the identification of the ncMTOC are key findings that can help contribute to developing better treatment for patients experiencing these kinds of subfertility or infertility issues. As we observed round spermatids in *Plk4* and *Sas4* cKO mice, procedures such as round spermatid injection (ROSI) or other reproductive interventions could be used to overcome infertility caused by centriole biogenesis defects (Tanaka and Watanabe, 2023).

## Conclusion

We have demonstrated that PLK4 is required for centriole duplication and maturation during mammalian spermatogenesis.

Despite a lack of centrioles in *Plk4* cKO spermatocytes, these cells successfully complete chromosome segregation by utilizing a ncMTOC mechanism. The exact mechanism by which PLK4 is regulated during spermatogenesis remains unknown. In mitotic cells, PLK4 is regulated by undergoing autophosphorylation and proteasomal degradation in order to prevent centriole overduplication. However, PLK4 is present at the centrosome throughout spermatogenesis, suggesting that PLK4 is regulated differently during meiosis. We hypothesize that PLK4 is regulated through functional inhibition during spermatogenesis. When PLK4 was overexpressed in spermatocytes we observed centriole overduplication. This indicates that excess PLK4 overcomes this predicted canonical inhibitory mechanism. Future research aimed at determining how PLK4 and, thus, centriole duplication, is regulated during mammalian spermatogenesis will be a critical next step.

## Methods

### Ethics statement

All mice were bred at Johns Hopkins University (JHU, Baltimore, MD), the Uniformed Services University of Health Sciences (USUHS, Bethesda, MD), and Universidad Autónoma de Madrid (UAM, Madrid, Spain). Protocols for their care and use were approved by the Institutional Animal Care and Use Committees (IACUC) of JHU, USUHS, and UAM. All research was performed in accordance with the National Institutes of Health and U.S. Department of Agriculture (JHU and USUHS) and European Union (UAM) guidelines. Studies involving deidentified donor testes tissues have been reviewed and designated by Johns Hopkins University Bloomberg School of Public Health and USUHS IRBs as "not human subjects research" (IRB No.: 00006700 and DBS.2024.685) and approved for use by The USU Human Anatomical Materials Review Committee (HAMRC; #71-22-034).

### Mice

Mice harboring the *Sas4* cKO allele (C57BL/6N; *Cenpj*^tm1a(EUCOMM)Wtsi) and mice harboring a *Plk1* cKO allele (C57BL/6N; Plk1 tm1c(EU-COMM)Hmgu) were previously described (Bazzi and Anderson, 2014; Phan et al, 2021; Little and Jordan, 2020; Wellard et al, 2021).

 

Within the *Sas4* allele are two loxP sites flanking the 5th exon of *Sas4* (McIntyre et al, 2012). Within the *Plk1* allele are two loxP sites flanking the 3rd exon of *Plk1*. The *Plk4* cKO allele (B6;SJL-*Plk4*^em1Ahol^/J) contains two loxP sites that flank the 5th exon of *Plk4* (LoMastro et al, 2022). Mice harboring the *Plk4*-mCherry OE allele were obtained from the Institute Curie, Paris. The *Plk4*-mCherry OE allele contains a floxed stop cassette upstream of the *Plk4*-mCherry sequence, preventing *Plk4*-mCherry expression prior to excision by Cre recombinase (Marthiens et al, 2013). Mice harboring the Centrin-GFP transgene (CB6-Tg(CAG-EGFP/CETN2)3-4Jgg/J) were obtained from the Jackson Laboratory (Higginbotham et al, 2004). Conditional mutation was achieved by crossing in a hemizygous Cre recombinase transgene that is under the control of a meiosis-specific promoter. The *Spo11* (Tg(Spo11-cre)1Rsw), *Stra8* (Tg(Stra8-iCre)1Reb/LguJ), and *Hspa2* (Tg(Hspa2-cre)1Eddy/J) promoters were used to drive Cre recombinase expression in this study (Lyndaker et al, 2013; Inselman et al, 2010; Sadate-Ngatchou et al, 2008).

## Oocyte harvesting and culture

Oocytes were collected as previously reported (Hwang et al, 2018b). Female mice were intraperitoneally injected with 5IU of pregnant mare serum gonadotropin (PMSG; Sigma) to prime ovarian follicles. After 44–48 h germinal vesicle-staged oocytes were harvested from the ovaries into M2 medium supplemented with 5% fetal bovine serum (FBS; Life Technologies), 3 mg/mL bovine serum albumin (BSA), and 10 mM milrinone (Sigma). Oocyte cumulus complexes were treated for 1–2 min in 300 IU/mL hyaluronidase (Sigma) in M2 medium supplemented with 3 mg/mL BSA to denude oocytes of the surrounding cumulus cells. Oocytes were then washed and cultured for 24 h in M2 media supplemented with 5% FBS and 3% BSA at 37 °C. Oocytes were visually assessed for progression to metaphase I and metaphase II. Metaphase II oocytes were collected for genotyping by intraperitoneally injecting female mice with 5IU of PMSG, followed by human chorionic gonadotropin (HCG; Sigma) 48 h later. Oocytes were collected from the ampulla, and ~20 denuded oocytes were lysed in 10 μL of PBND buffer that was heated for 30 min at 56 °C, 10 min at 95 °C, then brought to room temperature (Scavizzi et al, 2015).

## PCR genotyping

PCR genotyping was performed using AccuStart II PCR SuperMix (Quanta BioSciences). Primers used during this study are described in Appendix Table S1. PCR conditions: 94 °C for 2 min; 34–38 cycles of 94 °C for 20 s; 58 °C for 30 s; and 72 °C for 1 min, then a final extension of 72 °C for 10 min.

## Human testes

Deidentified decedent adult human donor testes (HT10, age 24) were obtained through the Washington Regional Transplant Consortium (WRTC, Annandale, VA, USA), which are now known as Infinite Legacy (Halethorpe, MD, USA).

## Tubule squash preparations

Mouse tubule squash preparations were performed as previously described (Wellard et al, 2018). Primary and secondary antibodies and dilutions used are presented in Appendix Table S2. Tubule squash preparations were mounted in Vectashield + DAPI (4', 6-diamidino-2-phenylindole) medium (Vector Laboratories). Full Z-stack captured images were utilized to manually identify centrioles, centrosomes, spindle morphology, and chromosome structure.

## Chromatin spread preparations

Mouse chromatin spread preparations were performed as previously described (Wellard et al, 2020). Primary and secondary antibodies and dilutions used are presented in Appendix Table S2. Chromatin spread preparations were immunolabeled and mounted in Vectashield + DAPI (Vector Laboratories). Full Z-stack captured images were utilized to manually identify chromosome structure.

## Inhibitor treatment of seminiferous tubules

Seminiferous tubules were prepared for inhibitor treatment as previously reported (Simerly et al, 2024). Testes from 24 to 27 dpp mice were removed into KRB enriched with 10 mM HEPES (eKRB). Tubules were transferred into 50-mL falcon tubes and allowed to settle for 5 min in eKRB at 34 °C. The supernatant was then decanted, and tubules were resuspended in 10 mL of 1 mg/ml collagenase IV in eKRB and agitated for 5 min at 37 °C. Following agitation, tubules were allowed to settle by gravity sedimentation and washed twice in eKRB. After the second wash, they were resuspended in eKRB containing either 10 μM of MLN8237 (AURKA inhibitor; Cayman Chemicals), 10 μM of ARRY-520 (KIF11 inhibitor, Cayman Chemicals), or 10 μM of DYNARRES-TIN (Dynein inhibitor, Cayman Chemicals) and incubated at 34 °C for 1–2 h. Following incubation, tubules underwent tubule squash preparation as described above.

## Culture of seminiferous tubules and centrinone-B treatment

Culture of seminiferous tubules was performed as previously described (Alfaro et al, 2021; Sato et al, 2011). Testes from adult C57BL/6 (WT) mice were removed, detunicated, and fragments of seminiferous tubules were cultured at 34 °C in an atmosphere with 5% $CO_2$ onto agarose gel half-soaked in MEMα culture medium (Gibco A10490-01) supplemented with KnockOut Serum Replacement (KOSR) (Gibco 10828-010), antibiotics (penicillin/streptomycin; Biochrom AG, A2213), and 1–100 μm of centrinone-B (Tocris, LCR-323). Controls were kept in MEMα culture medium without Centrinone-B. After 8 and 24 h, control and inhibitor-treated seminiferous tubules were subjected to the squashing technique described above.

## Histology and cryosectioning

For histological assessment, mouse testis tissue was fixed in bouins fixative (Ricca Chemical Company) prior to paraffin embedding. Serial sections, 5 μm thick, were mounted onto slides and stained with hematoxylin and eosin. For cryosectioning testis tissue was embedded in O.C.T. compound (Fisher) and frozen at −80 °C. Serial sections, 16 μm thick, were cut using Cryo3 (Sakura Tissue-Tek, Torrance,

CA, USA) and mounted onto Selectfrost Adhesion microscope slides (Fisher). Slides were then fixed with 2.5% formalin in PBS for 10 min at room temperature and washed in PBS. Slides were then immunolabeled as previously described (Atkins et al, 2020) with primary and secondary antibodies (see Appendix Table S2).

## Epididymal sperm count

Mouse epididymides (one per mouse) were dissected and placed into PBS, then cut into several smaller pieces and incubated for 30 min at 30 °C to allow time for sperm to be released from the tissue. The sperm-containing solution was then transferred to a fresh tube, separating released sperm from epididymal tissue, and diluted as necessary before counting sperm with a hemocytometer.

## Microscope image acquisition

Tubule squash and chromatin spread images were captured using a Zeiss CellObserver Z1 linked to an ORCA-Flash 4.0 CMOS camera (Hamamatsu). Histology and cryo-sectioning images were captured using a Keyence BZ-800 and BZ-X800 Viewer and Analyzer software. Images were processed using ZEN 2012 blue edition imaging software (Zeiss) or BZ-X800 Viewer and Analyzer software (Keyence). Images were analyzed using the Zeiss ZEN 2012 blue edition image software, and Photoshop (Adobe) was used to prepare the figure images.

## Microtubule length and width measurements

Spindle measurements were made using ZEN 2012 blue edition imaging software (Zeiss). The spindle length was obtained by measuring from where microtubule nucleation began at an MTOC to the aligned chromosomes on the metaphase I spindle. The width of the spindle was measured proximal to aligned chromosomes on the metaphase I spindle. The width and length of both spindles of a metaphase I spermatocyte were measured and reported as a ratio of A:B, with A being the smaller of the two length or width measurements (Fig. 6B,D).

## Data and statistical analysis

All graphics and statistical analyses were performed with GraphPad Prism 9.0 software. Each figure legend provides details of cell and animal counts and statistical analyses.

# Data availability

Request for data should be addressed to Philip Jordan (philip.jordan@usuhs.edu). Source data can be accessed through this link: https://www.ebi.ac.uk/biostudies/bioimages/studies/S-BIAD1185?key=a7d044f4-198e-4c8c-82ea-85edda37a38c.

The source data of this paper are collected in the following database record: biostudies:S-SCDT-10_1038-S44319-024-00187-6.

# Peer review information

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

## Acknowledgements

The authors thank Dr. Masatoshi Takeichi for the CAMSAP3 antibody, Dr. Hisham Bazzi for *Sas4* flox/flox mice (*Cenpj*^tm1c(EUCOMM)Wtsi), and Dr. Véronique Marthiens for mCherry-Plk4 mice (Tg(CAG-mCherry/Plk4)#Reba). The authors thank Infinite Legacy and the deidentified testis donor posthumously. This work was funded by NIGMS grant to PWJ (R01GM11755), NICHD grant to PWJ (R01HD114180), and training grant fellowships (NCI, NIH; CA009110; NICHD, NIH; F31HD111265) to MWS.

## Author contributions

**Marnie W Skinner**: Conceptualization; Data curation; Formal analysis; Funding acquisition; Investigation; Visualization; Methodology; Writing—original draft; Writing—review and editing. **Carter J Simington**: Data curation; Formal analysis; Validation; Investigation; Visualization; Writing—review and editing. **Pablo López-Jiménez**: Data curation; Formal analysis; Validation; Investigation; Visualization; Methodology; Writing—original draft; Writing—review and editing. **Kerstin A Baran**: Data curation; Formal analysis; Validation;

Visualization; Writing—review and editing. **Jingwen Xu**: Formal analysis; Validation; Investigation; Visualization; Writing—review and editing. **Yaron Dayani**: Formal analysis; Validation; Investigation; Visualization; Writing—review and editing. **Marina V Pryzhkova**: Data curation; Formal analysis; Validation; Investigation; Visualization; Methodology; Writing—review and editing. **Jesús Page**: Supervision; Writing—review and editing. **Rocío Gómez**: Conceptualization; Formal analysis; Supervision; Funding acquisition; Investigation; Methodology; Writing—original draft; Writing—review and editing. **Andrew J Holland**: Resources; Formal analysis; Investigation; Visualization; Writing—review and editing. **Philip W Jordan**: Conceptualization; Resources; Data curation; Formal analysis; Supervision; Funding acquisition; Investigation; Visualization; Methodology; Writing—original draft; Project administration; Writing—review and editing.

Source data underlying figure panels in this paper may have individual authorship assigned. Where available, figure panel/source data authorship is listed in the following database record: biostudies:S-SCDT-10_1038-S44319-024-00187-6.

## Disclosure and competing interests statement

Philip Jordan is on the scientific advisory board of Gameto, Inc. The remaining authors declare no competing interests.

# Expanded View Figures

**Figure EV1. Assessment of apoptosis during spermatocytes.**

(A–D) Cryosections (16 µm thick) of control and *Plk4* cKO mouse testis at 15 (A), 19 (B), 23 (C), and 27 (D) dpp, immunolabeled against SYCP3 (green), caspase-3 (red) and stained with DAPI (blue). Zoomed images of individual tubules are shown directly below the corresponding images. White arrows indicate an example of an apoptotic spermatocyte. Scale bar = 100 µm. (E) The *Plk1* cKO (*Plk1 flox/flox, Spo11-Cre tg/0*) mouse model has been reported previously to exhibit increased levels of apoptosis in primary spermatocytes and was used as a positive control (Wellard et al, 2021). Cryosections (16 µm thick) of control and *Plk1* cKO (34 dpp) mouse testis immunolabeled against SYCP3 (green), caspase-3 (red) and stained with DAPI (blue). Zoomed images of individual tubules are shown directly below the corresponding images. White arrows indicate an example of an apoptotic spermatocyte. Scale bar = 100 µm. (F) Quantification of apoptotic spermatocytes in control, *Plk4* cKO and *Plk1* cKO spermatocytes. Quantification was performed in ≥26 tubules. Error bars show mean ± SEM. $P$ values were obtained from two-tailed Student's $t$ test. n.s. (not significant) and ****$P < 0.0001$. $P$ value = <0.0001.

▶

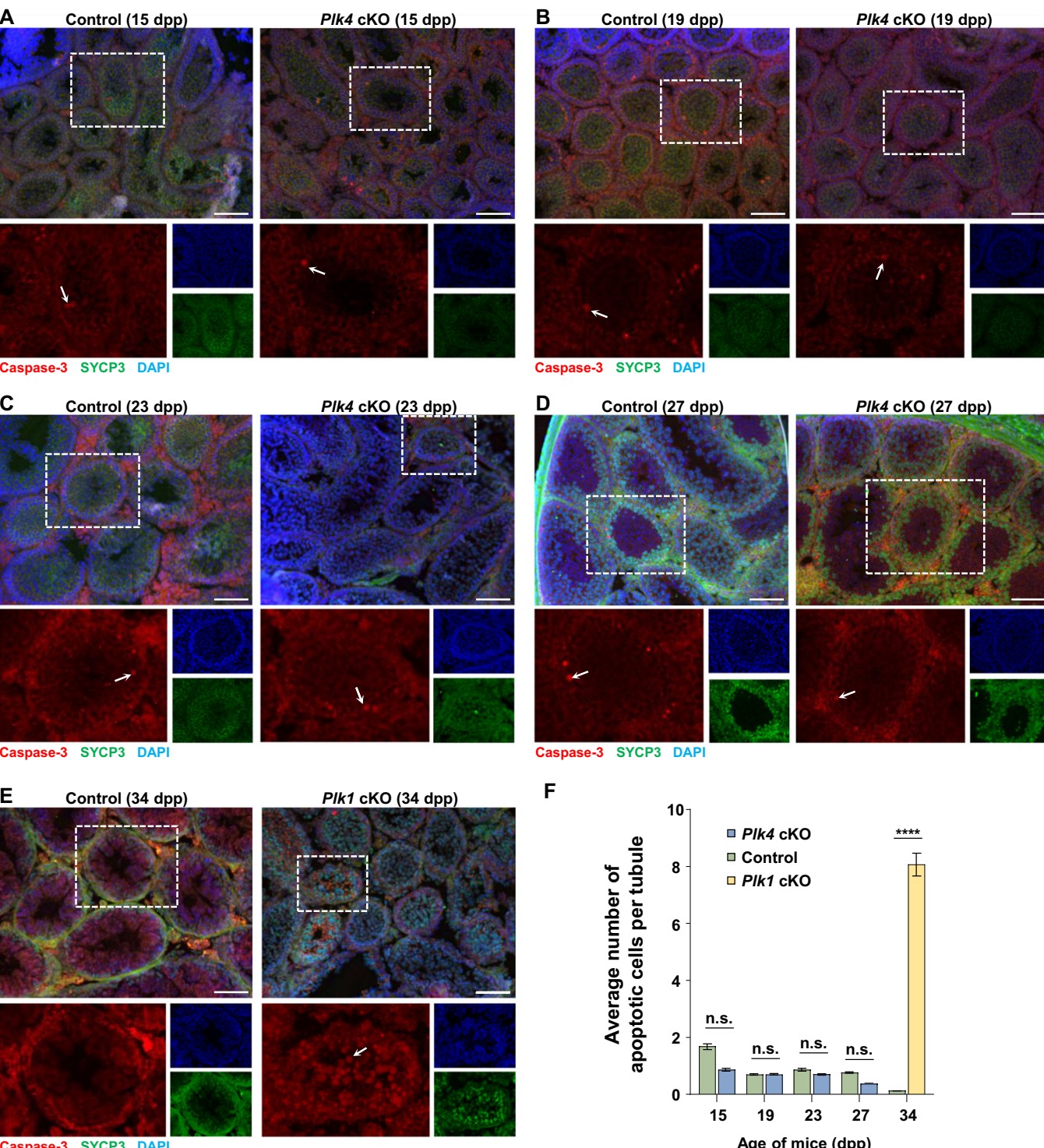

    This is a U.S. Government work and not under copyright protection in the US; foreign copyright protection may apply

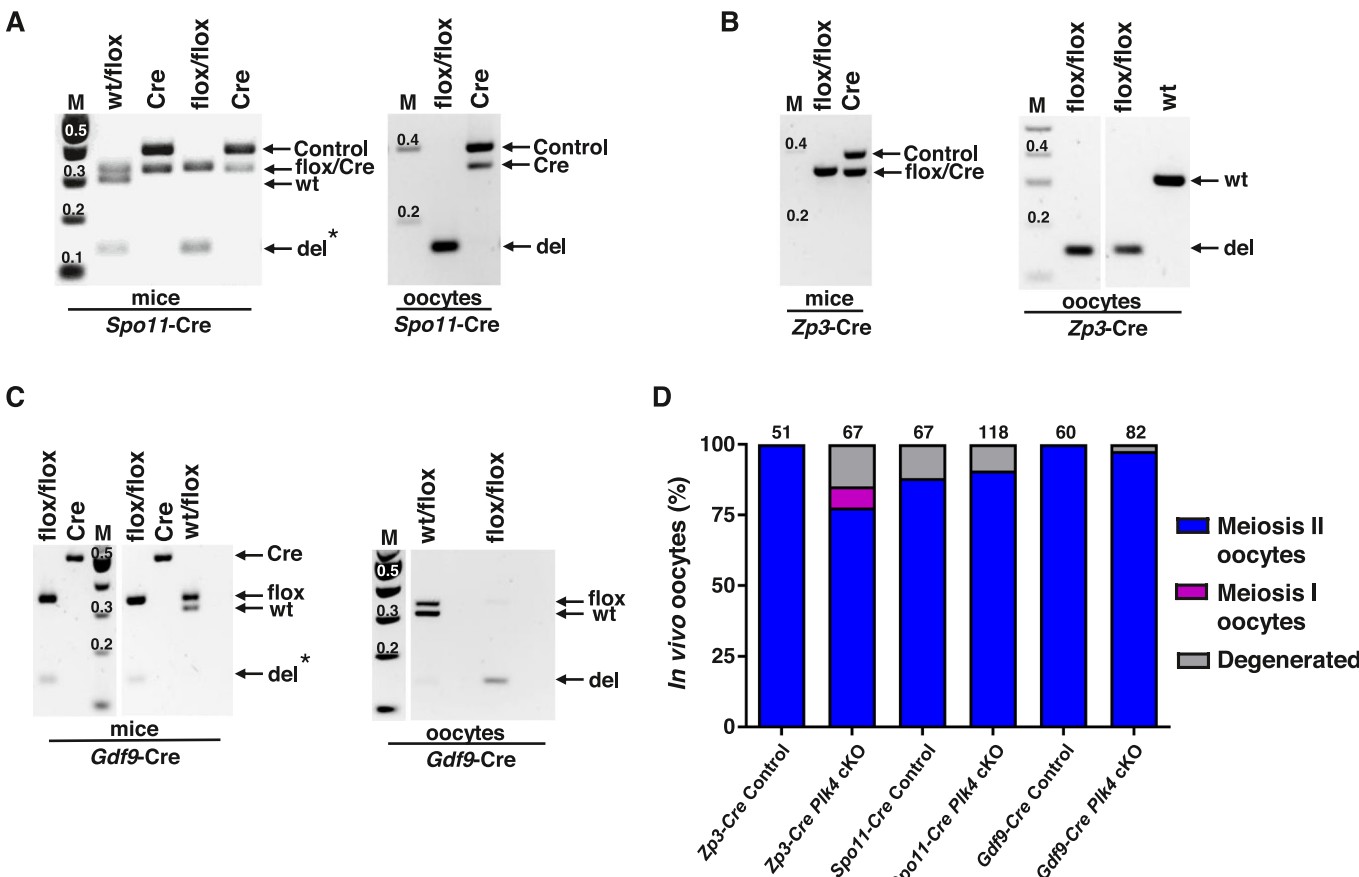

**Figure EV2. Assessment of *Plk4* cKO female mice.**

(A–C) PCR analysis of mouse and oocyte DNA showing the absence of the *flox* allele and presence of the del allele in oocytes after the expression of Cre recombinase driven by the *Spo11* (A), *Zp3* (B), and *Gdf9* (C) promoters. WT *Plk4* band 307 bp, flox floxed *Plk4* band 341 bp, del deletion band 143 bp, M DNA marker, kb kilobases, Cre Cre recombinase band 355 bp for *Spo11*-Cre and *Zp3*-Cre, and 500 bp for *Gdf9*-Cre, Cont internal control band for *Spo11*-Cre and *Zp3*-Cre recombinases 420 bp. *Non-tissue-specific Cre recombinase expression. (D) Oocytes from all *Plk4* cKO mice successfully progressed to MII stage in vivo. Mouse age 45-77 days. Oocytes from 3 superovulated mice were analyzed for each genotype. The total number of oocytes quantified from *Zp3*-cre control, *Zp3*-cre *Plk4* cKO, *Spo11*-cre control, *Spo11*-cre *Plk4* cKO, *Gdf9*-cre control and *Gdf9*-cre *Plk4* cKO mice were 51, 67, 67, 118, 60, and 82, respectively.

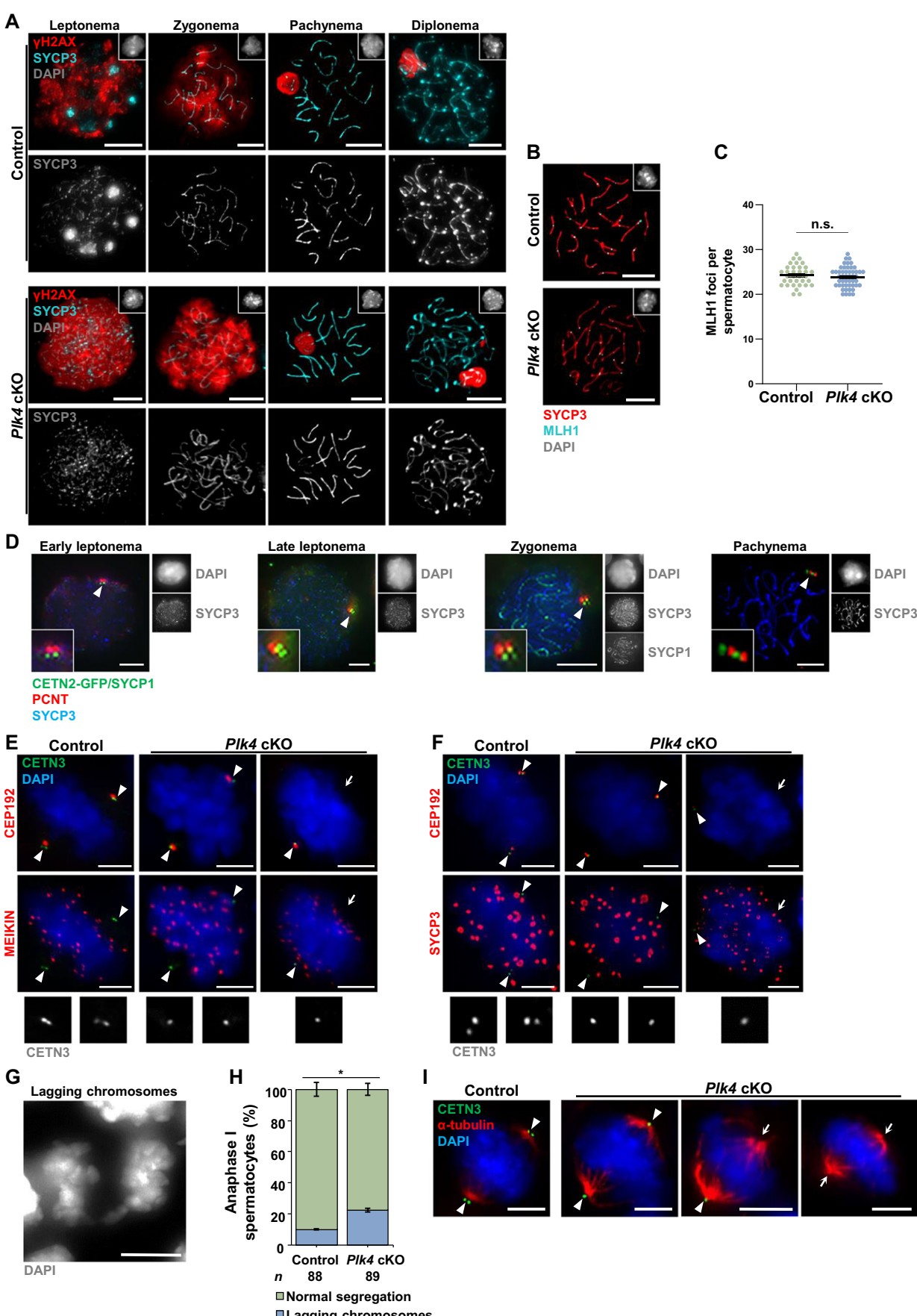

    This is a U.S. Government work and not under copyright protection in the US; foreign copyright protection may apply

**Figure EV3. Additional characterization of meiotic progression and centriole biogenesis during prophase I to metaphase II.**

(A) Prophase I control and *Plk4* cKO spermatocytes from 14 and 20 dpp mice were immunolabeled against SYCP3 (cyan), γ-tubulin (red), and stained with DAPI (gray inset in the upper right corner). Zoomed images of the centrioles are inset on the corresponding images. Scale bars = 10 μm. (B) Representative images of mid-prophase I spermatocytes in 20 dpp control and *Plk4* cKO mice immunolabeled against SYCP3 (red), MLH1 (cyan), and stained with DAPI (gray inset in the upper right corner). Scale bars = 10 μm. (C) Quantification of MLH1 foci observed along SYCP3 stretches during pachynema in both control and *Plk4* cKO spermatocytes. Immunolabeling was performed on 3 biological replicates with ≥ 10 spermatocytes quantified per replicate. The total number of cells quantified for control, and *Plk4* cKO mice were 34 and 49, respectively. *P* value = 0.4386. (D) Representative images of early leptotene stage, late leptotene stage, zygotene stage, and pachytene stage control spermatocytes expressing CETN2-GFP (green) from 10 dpp mice immunolabeled against SYCP1 (green), PCNT (red), SYCP3 (blue), and stained with DAPI (gray outset). The white arrowheads indicate the centrosome. Individual gray outset channel of SYCP3 and SYCP1 are displayed to the right of the corresponding images. Zoomed images of the centrioles and inset on corresponding images. Scale bars = 5 μm. (E) Characterization of metaphase I compared to metaphase II spermatocytes by observing DNA size (stained with DAPI) was confirmed with MEIKIN and SYCP3 immunostaining. Both SYCP3 and MEIKIN staining are present in metaphase I spermatocytes, but no longer in metaphase II spermatocytes (Kim et al, 2015). Representative image of metaphase I control and *Plk4* cKO spermatocytes from 27 dpp mice immunolabeled against CETN3 (green), MEIKIN (red) and stained with DAPI (blue). The white arrowheads indicate the centrosome. The white arrows indicate the ncMTOC. Zoomed images of the centrioles (gray) are displayed below the corresponding images. Scale bars = 5 μm. (F) Representative image of metaphase I control and *Plk4* cKO spermatocytes harboring the CETN2-GFP transgene (green) from 27 dpp mice immunolabeled against SYCP3 (red) and stained with DAPI (blue). The white arrowheads indicate the centrosome. The white arrows indicate the ncMTOC. Zoomed images of the centrioles (gray) are displayed below the corresponding images. Scale bars = 5 μm. (G) Representative image of chromosome missegregation event during metaphase I stained with DAPI. Scale bars = 5 μm. (H) Quantification of lagging chromosomes in anaphase I spermatocytes. Immunolabeling was performed on 5 biological replicates with ≥ 10 spermatocytes quantified per replicate. The total number of cells quantified for control and *Plk4* cKO mice were 89 and 89, respectively. *P* value = 0.0278. (I) Representative images of metaphase II spermatocytes in 27 dpp control and *Plk4* cKO mice expressing CETN2-GFP (green) immunolabeled against α-tubulin (red) and stained with DAPI (blue). The white arrowheads indicate the centrosome. The white arrows indicate the ncMTOC. Scale bars = 5 μm. Data information: For all graphs (C, H), error bars show mean ± SEM. *P* values were obtained from two-tailed Student's *t* test. n.s (not significant), *$P < 0.05$.

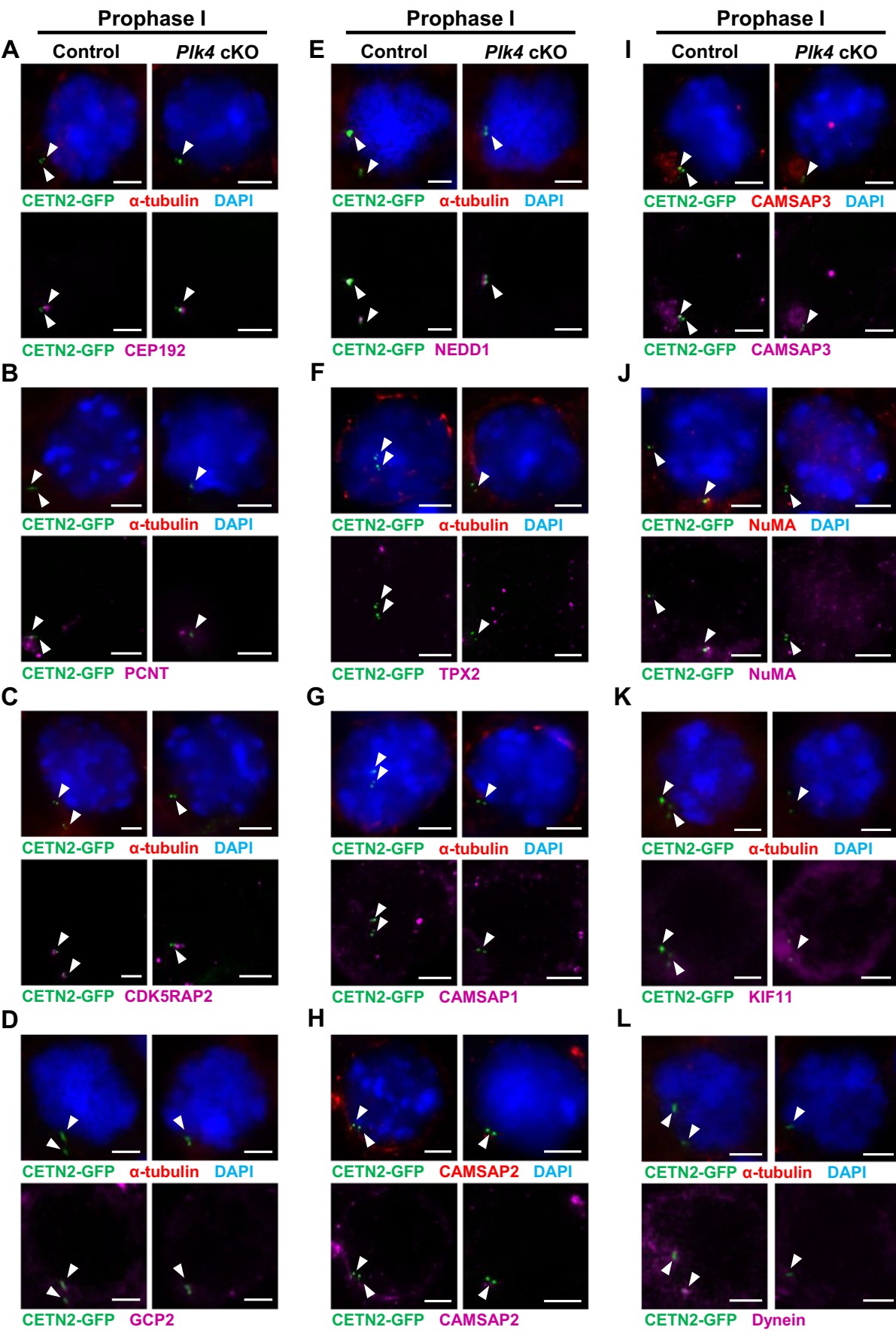

**Figure EV4. Assessment of microtubule-associated factors during prophase I.**

Representative image of prophase I (A–L) control and *Plk4* cKO spermatocytes from 24–27 dpp mice expressing CETN2-GFP (green) and stained with DAPI (blue). The white arrowheads indicate the centrosome. The white arrows indicate. Scale bars = 5 μm. (A) Immunolabeled against α-tubulin (red) and CEP192 (purple). (B) Immunolabeled against α-tubulin (red) and PCNT (purple). (C) Immunolabeled against α-tubulin (red) and CDK5RAP2 (purple). (D) Immunolabeled against α-tubulin (red) and GCP2 (purple). (E) Immunolabeled against α-tubulin (red) and NEDD1 (purple). (F) Immunolabeled against α-tubulin (red) and TPX2 (purple). (G) Immunolabeled against α-tubulin (red) and CAMSAP1 (purple). (H) Immunolabeled against CAMSAP2 (red and purple). (I) Immunolabeled against CAMSAP3 (red and purple). (J) Immunolabeled against NuMA (red and purple). (K) Immunolabeled against α-tubulin (red) and KIF11 (purple). (L) Immunolabeled against α-tubulin (red) and dynein (purple).

