## [Peer Review File · EMBO Reports]

Spermatocytes have the capacity to segregate chromosomes despite centriole duplication failure

Marnie Skinner, Carter Simington, Pablo López-Jiménez, Kerstin Baran, Jingwen Xu, Yaron Dayani, Marina Pryzhkova, Jesús Page, Rocio Gomez, Andrew Holland, and Philip Jordan

Corresponding author: Philip Jordan (pjordan8@jhu.edu)

Review Timeline:

Submission Date:	12th Dec 23
Editorial Decision:	23rd Jan 24
Revision Received:	22nd May 24
Editorial Decision:	12th Jun 24
Revision Received:	13th Jun 24
Accepted:	14th Jun 24

Editor: Esther Schnapp

Transaction Report:

Dear Phil,

Thank you for your patience while your manuscript was peer-reviewed at EMBO reports. We have now received the full set of referee reports that is pasted below.

As you will see, while referee 3 is concerned about the limited novelty of the findings, both referees 1 and 2 acknowledge that the data are interesting and a good fit for our journal. However, we also agree with referee 1 that some more data/insight into how spindle pole formation or microtubule nucleation occurs at centriole-free poles should be provided. We think that all referee suggestions are good and should be addressed. Please let me know in case you disagree, and we can discuss the exact revision requirements further, also in a video chat, if you like.

I would thus like to invite you to revise your manuscript with the understanding that the referee concerns must be fully addressed and their suggestions taken on board. Please address all referee concerns in a complete point-by-point response. Acceptance of the manuscript will depend on a positive outcome of a second round of review. It is EMBO reports policy to allow a single round of major revision only and acceptance or rejection of the manuscript will therefore depend on the completeness of your responses included in the next, final version of the manuscript.

We realize that it is difficult to revise to a specific deadline. In the interest of protecting the conceptual advance provided by the work, we recommend a revision within 3 months (24th Apr 2024). Please discuss the revision progress ahead of this time with the editor if you require more time to complete the revisions.

- 1) A data availability section providing access to data deposited in public databases is missing. If you have not deposited any data, please add a sentence to the data availability section that explains that.
- 2) Your manuscript contains statistics and error bars based on $n=2$. Please use scatter blots in these cases. No statistics should be calculated if $n=2$.

3) We replaced Supplementary Information with Expanded View (EV) Figures and Tables that are collapsible/expandable online. A maximum of 5 EV Figures can be typeset. EV Figures should be cited as 'Figure EV1, Figure EV2' etc... in the text and their respective legends should be included in the main text after the legends of regular figures.

5) a complete author checklist, which you can download from our author guidelines <https://www.embopress.org/page/journal/14693178/authorguide>. Please insert information in the checklist that is also reflected in the manuscript. The completed author checklist will also be part of the RPF.

6) Please note that all corresponding authors are required to supply an ORCID ID for their name upon submission of a revised manuscript (<https://orcid.org/>). Please find instructions on how to link your ORCID ID to your account in our manuscript tracking system in our Author guidelines <https://www.embopress.org/page/journal/14693178/authorguide#authorshipguidelines>

I look forward to seeing a revised form of your manuscript when it is ready.

Referee #1:

Skinner et al. hand their work on "Spermatocytes have the capacity to segregate chromosomes despite centriole duplication failure" that focuses on the analysis of Plk4 functions in spermatogenesis of mammals (mice/human).

They first describe the localization of Plk4 in human testes stating that Plk4 does not undergo degradation between the meiotic divisions while still being "in check" for centriole duplication. Overexpression of Plk4 overrides the "check" and leads to centriole overduplication. In turn, pharmacological inhibition of Plk4 using centrinone strongly interferes with centriole duplication - like a conditional Plk4 ko in mice. Using the latter, the authors then show that, despite largely inhibited centriole duplication, meiotic divisions run surprisingly normal with regular bipolar meiotic spindles assembled. The authors rationalize that male spindle assembly must use an acentriolar mechanism, as a back-up, that allows centriole-free spindle pole assembly. The latter is suggested to not use centrosomal (PCM) proteins, including gamma tubulin, but rather works via CAMSAP1-3, TPX2 and NuMA. Finally, it is shown that the observed infertility in male Plk4 ko mice is a consequence of compromised spermatogenesis with defects in cilia formation and nuclear organization rather than the meiotic divisions.

Taken together, this work is certainly innovative and intriguing. It underlines that, as seen in many other systems, that the absence of centriole-based MTOCs still allows spindle assembly and function. The further observation that a centriole-independent pathway in male meiosis is independent of many centrosomal proteins but runs alternatively via CAMSAP, NuMA and TPX2, is, to my opinion, the most interesting part and justifies publication of the work in EMBO Reports. Having said this, the last point is also my main point of concern. The mechanism of the CAMSAP-NuMA-TPX2 mediated spindle pole formation, and possibly even nucleation in general in centriole-free poles, is not well characterized. While I do concede that detailed mechanistical insights may go beyond the scope of this paper, I feel that some more information is required here. Experiments to address this issue could include the following approaches:

1. Even though Dynein SEEMS to be absent from the non-centriolar pole (I find it actually hard to draw this conclusion), its inhibition is not addressed. It would be rather straight-forward to use Dynein inhibitors to find out if this leaves the centriole-free spindle pole organization intact, while it may still affect pole organization in e.g. female meiosis when using the same concentration.
2. Eg5/Kif11 has not been tested either. Mostly being at interpolar MT, it may still be required at poles or in the vicinity of poles to organize them. It is certainly required for complete centriole-free female meiosis. The same as with Dynein: one could use inhibitors to test the role of Kif11 in this unusual centriole-free pole formation, or spindle assembly with very different pole organization(s).
3. TPX2 is found tightly associated with Aurora A kinase in mitosis. The latter seems not to localize (or could not be detected) in spindle poles in male meiosis, while TPX2 certainly does. Again, inhibitors may be used to double check if kinase activity is required for centriole-free pole formation.
4. The statement of missing gamma-tubulin is rather weak. IFs of controls are visible but not strong. Nothing is seen along spindle MT. The negative argument loses impact when the (positive) control is already hard to see. Higher signals and a second gamma-TuRC marker may help to make this point more convincing.

Minor issues:

First paragraph in results: PLK4 present in one or two spots during meiotic divisions; it should be stated clearly how this compares to mitotic divisions and what would be the theoretical options for meiosis.

Centrinone-Inhibitor concentration of a factor of two below what can be tolerated in the system is a challenging condition and may easily cause pleiotropic side effects. This should be discussed.

Fig. 5: The authors should comment on why they think the ko phenotype is only largely but not completely penetrant; is it e.g. a timing of ko issue?

Fig. 6: What does it mean that the "spindle pole was reduced both in length and width"? Does this rather mean spindle hemisphere? The legend of Fig. 6 says "microtubule size"? It may be integrated into the cartoon what is quantified to avoid any misunderstanding of what is measured here.

Referee #2:

This is a well-executed study that investigates the role of the kinase PLK4, the master regulator of centriole duplication, during spermatogenesis. Using mouse models and in vitro culture the authors show that PLK4 is present at centrosomes throughout

meiotic cycles raising the question of how centriole duplication is restricted to occur only once per cycle. While the mechanism of how this is achieved remains to be determined, using overexpression, conditional knockout and chemical inhibition the authors show that this regulation can be overcome to induce extra or fewer centrioles in spermatocytes, similar to what has been observed in mitotic systems. Interestingly, cells that contain only one centriole can still form bipolar spindles (with one acentriolar pole) and progress through meiotic divisions. Perhaps more surprisingly, the acentriolar pole does not contain any of the proteins found at acentriolar poles in mitotic systems such as gamma-tubulin or CDK5RAP2. Instead, the authors identify TPX2, NUMA and CAMSAP proteins to localize to these poles, suggesting they may be involved in acentriolar pole formation and/or bipolar spindle assembly. While PLK4 is not essential for meiotic divisions, it is essential for sperm flagellum formation and fertility.

Overall the data is of high quality and supports the conclusions. I only have a few minor comments that should be addressed before publication.

- 1) Abstract (lines 23-27) and discussion (first paragraph) claim discovery of a new acentriolar mechanism of spindle assembly, but the mechanism has not been investigated. For example, where are MTs nucleated? At the remaining centrosome? At the chromatin? Is gamma-tubulin required? It likely is, despite no permanent localization at the acentriolar pole. Do monopolar spindles form first? Similarly, the involvement of TPX2, NUMA, and CAMSAPs, despite their localization to the acentriolar poles, has not been investigated. These issues could be discussed in more detail, but I suggest to not refer to "mechanism" and rephrase the corresponding sections.
- 2) Lines 269-272: I would explicitly state the increase in percentage with only one centriole to make the point of centriole loss clearer.
- 3) Earlier work, coauthored by one of the authors (Marjanovic et al. 2015, Nat Comms, DOI: 10.1038/ncomms8676) suggested a requirement for centrioles in meiotic recombination. I assume the reason why this is not observed in the current study is the timing of centriole loss. It seems that the presence of non-duplicated centrioles during meiosis I (as in the current study) is sufficient for providing this function. The earlier work could be cited and this could be discussed.
- 4) Suppl. Fig. 9: Some stainings seem to be triple stainings plus DAPI with alpha-tub shown in red. Others lack alpha-tub staining but still show both red and purple coloring for the single channel of the spindle associated proteins. This is redundant and confusing. I suggest to show only a single color (purple) as for the triple stainings. Similarly, for consistency, I would use the same color scheme (purple for spindle associated proteins) in Fig. 7B.
- 5) The presentation in Fig. 1A-D is confusing. What is shown in grey in the insets in the upper right corners? In general the insets should be magnified versions of the small white squares outlined in the main image. The PLK4 channels below the main image don't seem to be exactly the same as in the main image, at least they seem to show less detail. They also seem to be a bit redundant unless they provide visible detail not seen in the main image. The magnified squares showing the centrosome staining could be bigger.

Referee #3:

The manuscript by Skinner et al investigates the role of Plk4 during spermatogenesis using mouse models to overexpress or conditionally knock out Plk4. The manuscript is well written and the great majority of experiments are very well executed and nicely quantified. Having said that, this reviewer struggled to identify the novel findings in this work. The authors show that Plk4 is important for centriole duplication, and that its overexpression results in over duplication. These are well established in the field for nearly 20 years from many systems. The most novel finding is the non-centriolar localization of Plk4; however, the manuscript does not investigate this aspect and does not capitalize on the main strength of this system over cultured systems. Finally, the claim that they discovered a novel mechanism of spindle assembly that ensures chromosome segregation when centriole duplication fails is simply not true. What the authors found is that factors known to be involved in normal pole formation, and in other non-centrosome conditions, is also used in this context. Overall, the work is well done, but the novelty escapes me.

Major comments

1. The authors completely dismiss all work done in *Drosophila* spermatogenesis. Work from nearly 15 years ago from Bettencourt Dias on SAK, work by Callani on Sas-4, work by Avidor-Reiss on Plk4 that shows non-centriole duplication role for Plk4 in spermatogenesis, and many other studies were not mentioned.
2. Lines 270-281 include the idea that loss of Plk4 results in failed centriole maturation (becoming a mother), which results in degradation of the "immature parent centriole". This loss is presumably due the inability to become an independent MTOC, leading to centriole instability and disassembly (they reference 58 and 59). While the previous work might have shown this, I do not see how this conclusion is made based on the data presented here. Do the authors see immature parents that then are degraded? How is this shown? The presence of 1 centriole in 20% of plk4 cKO compared to loss of sas4 seems insufficient to make this conclusion. Given this is not a critical aspect of the paper, I suggest removing this section.

3. The section on TPX2, CAMSAP1-3 and NuMA is presented as if there was a major discovery or at least a "novel observation". This is simply not true. One can find extensive literature in many systems that show this is true in a-centrosomal conditions.

Minor comments

Line 130 - refence needed for Hspa2 expression pattern.

Line 157 - The argument for the use of higher concentration of centrinone is based on media diffusion. It could also be a result of a difference in biology between systems. For example, there might be higher levels or more stable amounts of Plk4

Figure 3a - are there typically more sperm in centrinone treated conditions as shown?

Figure 3C - should include untreated control cartoon to show WT for the non-expert

Figure 3c-e - why show 8hr treatment here instead of 24 hr? Most the experiments subsequently are 24 hr

Figure 4g - can the authors show examples of abnormal sperm

Figure 7b - the chart is unnecessarily confusion. It seems more suited for a review.

Figure 10 - Do spermatids with centrioles form flagella? How frequently were defects seen?

Firstly, we would like to thank the reviewers for their time and expert critique of our manuscript. We have addressed all comments. We believe that our manuscript has been made much stronger thanks to addressing your feedback and hope that you will deem it suitable for acceptance at EMBO reports.

Our responses are highlighted in yellow, and the references to the text and figures are highlighted in green.

Referee #1:

Skinner et al. hand their work on "Spermatocytes have the capacity to segregate chromosomes despite centriole duplication failure" that focuses on the analysis of Plk4 functions in spermatogenesis of mammals (mice/human).

They first describe the localization of Plk4 in human testes stating that Plk4 does not undergo degradation between the meiotic divisions while still being "in check" for centriole duplication. Overexpression of Plk4 overrides the "check" and leads to centriole overduplication. In turn, pharmacological inhibition of Plk4 using centrinone strongly interferes with centriole duplication - like a conditional Plk4 ko in mice. Using the latter, the authors then show that, despite largely inhibited centriole duplication, meiotic divisions run surprisingly normal with regular bipolar meiotic spindles assembled. The authors rationalize that male spindle assembly must use an acentriolar mechanism, as a back-up, that allows centriole-free spindle pole assembly. The latter is suggested to not use centrosomal (PCM) proteins, including gamma tubulin, but rather works via CAMSAP1-3, TPX2 and NuMA. Finally, it is shown that the observed infertility in male Plk4 ko mice is a consequence of compromised spermatogenesis with defects in cilia formation and nuclear organization rather than the meiotic divisions.

Taken together, this work is certainly innovative and intriguing. It underlines that, as seen in many other systems, that the absence of centriole-based MTOCs still allows spindle assembly and function. The further observation that a centriole-independent pathway in male meiosis is independent of many centrosomal proteins but runs alternatively via CAMSAP, NuMA and TPX2, is, to my opinion, the most interesting part and justifies publication of the work in EMBO Reports. Having said this, the last point is also my main point of concern. The mechanism of the CAMSAP-NuMA-TPX2 mediated spindle pole formation, and possibly even nucleation in general in centriole-free poles, is not well characterized. While I do concede that detailed mechanistical insights may go beyond the scope of this paper, I feel that some more information is required here. Experiments to address this issue could include the following approaches:

We appreciate the request to include additional mechanistic insights in our study. To this end, we performed experiments to determine the relationship between PLK1 and PLK4. In a prior publication, we determined that conditional heterozygous mutation of *Plk1* (*Plk1* cHet) in spermatocytes results in centriole overduplication in prophase spermatocytes (Wellard et al. 2021). Therefore, we assessed the compound mutation of *Plk1* cHet; *Plk4* cKO to determine whether PLK4 function was upstream or downstream of PLK1 function regarding centriole duplication. We report that *Plk1* cHet; *Plk4* cKO spermatocytes undergo centriole duplication failure, as observed in *Plk4* cKO spermatocytes, indicating that PLK4 function is upstream to that of PLK1. Please see lines 358-368 and Figure 5K-N.

We also address additional suggestions to assess Dynein, Aurora A kinase, and KIF11/Eg5, see below.

1. Even though Dynein SEEMS to be absent from the non-centriolar pole (I find it actually hard to draw this conclusion), its inhibition is not addressed. It would be rather straight-forward to use Dynein

inhibitors to find out if this leaves the centriole-free spindle pole organization intact, while it may still affect pole organization in e.g. female meiosis when using the same concentration.

We assessed the localization of Dynein in *Plk4* cKO and control spermatocytes at meiosis I using newly acquired antibodies (See Appendix Table S2). We determined that Dynein does not localize to the ncMTOC in *Plk4* cKO spermatocytes. Please see lines 405-412 and Figure 8G and EV4L-N.

Thank you for your suggestion regarding inhibiting dynein to determine whether it is important for the formation/stability of the acentrosomal pole. Using isolated tubules from control and *Plk4* cKO mice undergoing the first wave of spermatogenesis, we inhibited Dynein with Dynarrestin. We determined that despite a slight disturbance to metaphase plate alignment, *Plk4* cKO spermatocytes harboring a ncMTOC continued to form bipolar spindles when treated with Dynarrestin. Please see lines 439-449 and Figure 8D-G.

2. Eg5/Kif11 has not been tested either. Mostly being at interpolar MT, it may still be required at poles or in the vicinity of poles to organize them. It is certainly required for complete centriole-free female meiosis. The same as with Dynein: one could use inhibitors to test the role of Kif11 in this unusual centriole-free pole formation, or spindle assembly with very different pole organization(s).

We assessed the localization of KIF11/Eg5 in *Plk4* cKO and control spermatocytes at meiosis I using a newly acquired antibody (See Appendix Table S2). We determined that KIF11 does localize to the ncMTOC. Please see lines 413-425 and Figure 7B.

Thank you for your suggestion regarding inhibiting KIF11/Eg5 to determine whether it is important for the formation/stability of the acentrosomal pole. Using isolated tubules from control and *Plk4* cKO mice undergoing the first wave of spermatogenesis, we inhibited KIF11/Eg5 with ARRY-520. We determined that KIF11 is required for maintaining a bipolar spindle in *Plk4* cKO spermatocytes that harbor a single centriole. The maintenance of a bipolar spindle was also affected in the control spermatocytes. Taken together, this suggests that KIF11 is critical for both centrosomal and non-centrosomal bipolar spindle formation in mammalian spermatocytes. Please see lines 450-458 and Figure 8H-J.

3. TPX2 is found tightly associated with Aurora A kinase in mitosis. The latter seems not to localize (or could not be detected) in spindle poles in male meiosis, while TPX2 certainly does. Again, inhibitors may be used to double check if kinase activity is required for centriole-free pole formation.

We previously reported AURKA localization during spermatogenesis, and the signal obtained for AURKA was specific (as determined by *Aurka* cKO), but was also punctate along the spindle (Wellard et al. 2021). This pattern was much less robust in signal intensity compared to TPX2. We tried many preparations to improve our immunostaining of AURKA using multiple antibodies (Thermo, MA5-15803; Abclonal, A2121; Cell signaling, 2914). We determined that AURKA cannot be detected at the ncMTOC in *Plk4* cKO spermatocytes. We have commented further on this issue. See lines 426-429.

Thank you for your suggestion regarding inhibiting Aurora A kinase to determine whether it is important for the formation/stability of the acentrosomal pole. Using isolated tubules from control and *Plk4* cKO mice undergoing the first wave of spermatogenesis, we inhibited Aurora A kinase with MLN8237. We determined that inhibition of AURKA does not affect TPX2 localization of bipolar spindle formation in *Plk4* cKO spermatocytes harboring a ncMTOC. As a control, we assessed the localization of another known AURKA binding partner and substrate, TACC3. Inhibition of AURKA led to the disruption of TACC3 localization to the spindle poles in mouse spermatocytes at metaphase I, as previously reported (Fig. 8A-C) (Simerly et al. 2024). Please see lines 426-438 and Figure 8A-C.

4. The statement of missing gamma-tubulin is rather weak. IFs of controls are visible but not strong. Nothing is seen along spindle MT. The negative argument loses impact when the (positive) control is already hard to see. Higher signals and a second gamma-TuRC marker may help to make this point more convincing.

We have assessed gamma-tubulin localization previously (Wellard et al. 2021). The localization of gamma-tubulin during spermatogenesis is more punctate compared to what is observed in mitotically dividing cells (Wellard et al. 2021). To further our assessment of gamma-tubulin, we assessed additional components of the gamma-TuRC complex (GCP2 and GCP4) using additional antibodies (See Appendix Table S2). We determined that both GCP2 and GCP4 do not localize to the ncMTOC in *Plk4* cKO spermatocytes. Please see lines 381-390, lines 561-563, as well as Figure 6A and EV4S-T.

Minor issues:

First paragraph in results: PLK4 present in one or two spots during meiotic divisions; it should be stated clearly how this compares to mitotic divisions and what would be the theoretical options for meiosis.

We have added this information in our revised manuscript. See lines 134-142.

Centrinone-Inhibitor concentration of a factor of two below what can be tolerated in the system is a challenging condition and may easily cause pleiotropic side effects. This should be discussed.

We have discussed this information in our revised manuscript. See lines 210-214.

Fig. 5: The authors should comment on why they think the ko phenotype is only largely but not completely penetrant; is it e.g. a timing of ko issue?

We have discussed this information in our revised manuscript. See lines 327-331.

Fig. 6: What does it mean that the "spindle pole was reduced both in length and width"? Does this rather mean spindle hemisphere? The legend of Fig. 6 says "microtubule size"? It may be integrated into the cartoon what is quantified to avoid any misunderstanding of what is measured here.

We have revised this information in the figure legend and represented it in the cartoons. See lines 391-329, lines 732-738, and Figure 6B.

Referee #2:

This is a well-executed study that investigates the role of the kinase PLK4, the master regulator of centriole duplication, during spermatogenesis. Using mouse models and in vitro culture the authors show that PLK4 is present at centrosomes throughout meiotic cycles raising the question of how centriole duplication is restricted to occur only once per cycle. While the mechanism of how this is achieved remains to be determined, using overexpression, conditional knockout and chemical inhibition the authors show that this regulation can be overcome to induce extra or fewer centrioles in spermatocytes, similar to what has been observed in mitotic systems. Interestingly, cells that contain only one centriole can still form bipolar spindles (with one acentriolar pole) and progress through meiotic divisions. Perhaps more surprisingly, the acentriolar pole does not contain any of the proteins

found at acentriolar poles in mitotic systems such as gamma-tubulin or CDK5RAP2. Instead, the authors identify TPX2, NUMA and CAMSAP proteins to localize to these poles, suggesting they may be involved in acentriolar pole formation and/or bipolar spindle assembly. While PLK4 is not essential for meiotic divisions, it is essential for sperm flagellum formation and fertility.

Overall the data is of high quality and supports the conclusions. I only have a few minor comments that should be addressed before publication.

1) Abstract (lines 23-27) and discussion (first paragraph) claim discovery of a new acentriolar mechanism of spindle assembly, but the mechanism has not been investigated. For example, where are MTs nucleated? At the remaining centrosome? At the chromatin? Is gamma-tubulin required? It likely is, despite no permanent localization at the acentriolar pole. Do monopolar spindles form first? Similarly, the involvement of TPX2, NUMA, and CAMSAPs, despite their localization to the acentriolar poles, has not been investigated. These issues could be discussed in more detail, but I suggest to not refer to "mechanism" and rephrase the corresponding sections.

Thank you for your guidance. We have rephrased the abstract and discussion to concur. See lines 23-26 and lines 510-511.

2) Lines 269-272: I would explicitly state the increase in percentage with only one centriole to make the point of centriole loss clearer.

We have included that statement. Please see lines 326-327.

3) Earlier work, coauthored by one of the authors (Marjanovic et al. 2015, Nat Comms, DOI: 10.1038/ncomms8676) suggested a requirement for centrioles in meiotic recombination. I assume the reason why this is not observed in the current study is the timing of centriole loss. It seems that the presence of non-duplicated centrioles during meiosis I (as in the current study) is sufficient for providing this function. The earlier work could be cited and this could be discussed.

We have included reference to this work and discussed the point that centrosome biogenesis may be linked with meiotic recombination. See lines 300-315.

While it is possible that centrosome biogenesis may affect meiotic recombination, we also point out that that assessment of the *Cep63* gene-trap allele serves as a model for Seckel syndrome. From our ongoing studies, a ciliopathy can significantly impact meiosis progression beyond the effects attributed solely to centriole duplication during zygotene. Additionally, knowledge of the timing of centrosome and centriole duplication in meiosis was limited back in 2015. This has significantly evolved with our two EMBO Reports publications in 2021 (Wellard et al. 2021 and Alfaro et al. 2021). The differences may also be explained by the fact that the *Cep63* gene-trap mutation was present within all mouse cells, whereas the *Plk4* cKO only occurred in primary spermatocytes. For instance, the *Cep63* gene-trap mutation leads to microcephaly (Marjanović et al, 2015), which could affect the regulation of the hypothalamus-pituitary gonadal axis (Cerbone et al, 2020).

4) Suppl. Fig. 9: Some staining's seem to be triple staining's plus DAPI with alpha-tub shown in red. Others lack alpha-tub staining but still show both red and purple coloring for the single channel of the spindle associated proteins. This is redundant and confusing. I suggest to show only a single color (purple) as for the triple stainings. Similarly, for consistency, I would use the same color scheme (purple for spindle associated proteins) in Fig. 7B.

We have rearranged this figure and added additional factors that were assessed. We have kept the figure panel layout and coloring consistent with one another. This figure is now titled Figure EV4.

5) The presentation in Fig. 1A-D is confusing. What is shown in grey in the insets in the upper right corners? In general the insets should be magnified versions of the small white squares outlined in the main image. The PLK4 channels below the main image don't seem to be exactly the same as in the main image, at least they seem to show less detail. They also seem to be a bit redundant unless they provide visible detail not seen in the main image. The magnified squares showing the centrosome staining could be bigger.

We have edited Figure 1A-D. Please see Figure 1A-F.

Referee #3:

The manuscript by Skinner et al investigates the role of Plk4 during spermatogenesis using mouse models to overexpress or conditionally knock out Plk4. The manuscript is well written and the great majority of experiments are very well executed and nicely quantified. Having said that, this reviewer struggled to identify the novel findings in this work. The authors show that Plk4 is important for centriole duplication, and that its overexpression results in over duplication. These are well established in the field for nearly 20 years from many systems. The most novel finding is the non-centriolar localization of Plk4; however, the manuscript does not investigate this aspect and does not capitalize on the main strength of this system over cultured systems. Finally, the claim that they discovered a novel mechanism of spindle assembly that ensures chromosome segregation when centriole duplication fails is simply not true. What the authors found is that factors known to be involved in normal pole formation, and in other non-centrosome conditions, is also used in this context. Overall, the work is well done, but the novelty escapes me.

Major comments

1. The authors completely dismiss all work done in *Drosophila* spermatogenesis. Work from nearly 15 years ago from Bettencourt Dias on SAK, work by Callani on Sas-4, work by Avidor-Reiss on Plk4 that shows non-centriole duplication role for Plk4 in spermatogenesis, and many other studies were not mentioned.

We thank the reviewer for pointing out the need to discuss these important works. We have included paragraphs in our discussion that highlight these works and other studies to relate them to our newly reported findings in mammalian spermatogenesis. Please see lines 551-583.

2. Lines 270-281 include the idea that loss of Plk4 results in failed centriole maturation (becoming a mother), which results in degradation of the "immature parent centriole". This loss is presumably due the inability to become an independent MTOC, leading to centriole instability and disassembly (they reference 58 and 59). While the previous work might have shown this, I do not see how this conclusion is made based on the data presented here. Do the authors see immature parents that then are degraded? How is this shown? The presence of 1 centriole in 20% of plk4 cKO compared to loss of sas4 seems insufficient to make this conclusion. Given this is not a critical aspect of the paper, I suggest removing this section.

We have further bolstered this section by including additional data regarding the observation of spermatocytes with 2 centrioles becoming spermatocytes with 1 centriole. See lines 332-346. We respectfully disagree that removing it is necessary, given that we provide further details, prior work also supports this observation, and it is a scenario that does not occur when centriole duplication is prevented by lack of a structural component (SAS4) rather than the absence of the regulator of centriole duplication (PLK4).

3. The section on TPX2, CAMSAP1-3 and NuMA is presented as if there was a major discovery or at least a "novel observation". This is simply not true. One can find extensive literature in many systems that show this is true in a-centrosomal conditions.

We discuss prior observations regarding TPX2, CAMSAP1-3, and NuMA within the relevant results section (see lines 413-425) and the discussion section (see lines 510-530). We have removed the term "novel observation" and the word novel from the text and simply include them as observations.

Minor comments

Line 130 - reference needed for Hspa2 expression pattern.

Included. See line 177.

Line 157 - The argument for the use of higher concentration of centrinone is based on media diffusion. It could also be a result of a difference in biology between systems. For example, there might be higher levels or more stable amounts of Plk4

We have adjusted this accordingly. See lines 208-214.

Figure 3a - are there typically more sperm in centrinone treated conditions as shown?

No, the control and centrinone-B treated groups are from the same source of spermatocytes.

Therefore, there is no difference between them in regard to any cell stage of spermatogenesis. It is just a happenstance that there are more elongated spermatids in the field of view. There are more round spermatids in the control field of view.

Figure 3C - should include untreated control cartoon to show WT for the non-expert

We have included an untreated control cartoon. See Figure 3D.

Figure 3c-e - why show 8hr treatment here instead of 24 hr? Most the experiments subsequently are 24 hr

We have included both the 8 hr and 24 hr treatment figures. See Figure 3B-C and 3F-G.

Figure 4g - can the authors show examples of abnormal sperm

We assess the spermiogenesis abnormalities in detail in Figure 9A-H and show additional examples of abnormal sperm in Figure 9H-J.

Figure 7b - the chart is unnecessarily confusion. It seems more suited for a review.

We believe the reviewer is referring to Figure 7a. We respectfully disagree. As this is a minor comment and not noted by the other two reviewers, we would like to opt to keep this feature.

Figure 10 - Do spermatids with centrioles form flagella? How frequently were defects seen?

The Plk4 cKO spermatids never form a normal flagellum. The defects we see include the absence of flagella, the absence of δ -tubulin at the perinuclear ring structure, and misshapen spermatozoa heads characterized with DNA and acrosome staining. These defects are fully penetrant because flagella formation requires two centrioles to be inherited by the spermatid. This is necessary because one centriole becomes the proximal centriole that lies perpendicular to the base of the spermatid head, and the other becomes an atypical distal centriole. We have further elaborated on these observations.

See lines 480-504 and Figures 9E-J.

Dear Dr. Jordan,

Thank you for the submission of your revised manuscript. We have now received the enclosed reports from the referees that were asked to assess it. Referee 2 has one more suggestions that I would like you to incorporate before we can proceed with the official acceptance of your manuscript.

A few editorial requests will also need to be addressed:

- Please reduce the number of keywords to 5.
- Please rename the conflict of interest subheading to "Disclosure and Competing Interest Statement"
- Please correct the author name discrepancy: Rocío Gómez in the ms file versus Rocio Gomez Lencero in our online submission system.
- The author credits need to be removed from the ms file. All credits need to be entered during online ms submission.
- The Abbreviations section needs to be removed from the manuscript. Abbreviations should be defined in brackets after their first mention in the text, not in a list of abbreviations.
- Disclaimer section and Materials and correspondence should be removed from the ms.
- Materials and Methods needs to be called Methods.
- Figure EV4 runs over 3 pages, we don't allow that. All figure must fit onto one page. Please amend.
- Please note that the figure 3f-g is mislabeled as figure 3f in the manuscript. This needs to be rectified.
- Please note that the figure 8b is mislabeled as figure 8d in the data information section of the manuscript. This needs to be rectified.
- Please note that the exact p values are not provided in the legends of figures 2i-j, l; 3f-g; 4g, i; 5b, f, h, j, n; 6d; 9c-d; EV 1f; EV 3h. Please add the exact values.
- Please note that the scale bar is missing for figure 9b.
- Please note that the scale bar needs to be defined for figure EV 3g.

I would like to suggest a few minor changes to the abstract that needs to be written in present tense. Please also add information on the species used in this study to the abstract. Please let me know whether you agree with the following:

Centrosomes are the canonical microtubule organizing centers (MTOCs) of most mammalian cells, including spermatocytes. Centrosomes comprise a centriole pair within a structurally ordered and dynamic pericentriolar matrix (PCM). Unlike in mitosis, where centrioles duplicate once per cycle, centrioles undergo two rounds of duplication during spermatogenesis. The first duplication is during early prophase I, and the second is during interkinesis. Using mouse mutants and chemical inhibition, we here block centriole duplication during spermatogenesis and show that non-centrosomal MTOCs (ncMTOCs) can mediate chromosome segregation. This mechanism is different [OK?] from the acentriolar MTOCs that form bipolar spindles in oocytes, which require PCM components, including gamma-tubulin and CEP192. From an in-depth analysis, we identify six microtubule-associated proteins, TPX2, KIF11, NuMA, and CAMSAP1-3, that localize to the non-centrosomal MTOC. These factors contribute to a mechanism that ensures bipolar MTOC formation and chromosome segregation during spermatogenesis when centriole duplication fails. However, despite the successful completion of meiosis and round spermatid formation, centriole inheritance and PLK4 function are required for normal spermiogenesis and flagella assembly, which are critical for fertility.

EMBO press papers are accompanied online by A) a short (1-2 sentences) summary of the findings and their significance, B) 2-3 bullet points highlighting key results and C) a synopsis image that is exactly 550 pixels wide and 200-600 pixels high (the height is variable). The synopsis image should provide a sketch of the major findings, like a graphical abstract. Please note that text needs to be readable at the final size. Please send us this information along with the final manuscript.

I look forward to seeing a final version of your manuscript as soon as possible. Please use this link to submit your revision: <https://embo.msubmit.net/cgi-bin/main.plex>

Referee #1:

I stay with my previous evaluation that the work of Skinner et al. is innovative and interesting. Having revised their work, the authors further characterize a centriole-independent pathway in male meiosis that even does not require key centrosomal proteins but runs via CAMSAP, NuMA and TPX2.

A main point of criticism was that the negative conclusion of the pathway not requiring otherwise central spindle pole organizers such as Dynein, KIF11, Aurora A and g-TuRC was not well documented. Several approaches added to the manuscript (Fig. 5 and 8 in particular) document the effort to test these activities, or their localization, and either confirm (Dynein, Aurora A, g-TuRC) that they are not involved, or a requirement (KIFF11). Furthermore, the authors have addressed several minor points I had raised (see e.g. Fig. 6B). Taken together, to my opinion, the current version is significantly increased in clarity & quality to justify publication of the manuscript in EMBO R in its present form.

Referee #2:

The authors have addressed most of the concerns raised by the reviewers by both text changes and addition of new data.

Regarding referencing previous work in other systems (reviewer 3, point 1), the authors have now included a discussion of these studies. However, I strongly suggest to mention and cite some of the key findings from these other systems already in the introduction, to position the current study in the appropriate context. This is important for the non-expert reader to understand the premise for the current work and be able to assess the novelty of the findings.

Thank you for your reviews. Our responses are highlighted in yellow, and the references to the text and figures are highlighted in green.

Editorial Edits:

- Please reduce the number of keywords to 5.

We have adjusted this accordingly. See line 35.

- Please rename the conflict of interest subheading to "Disclosure and Competing Interest Statement"

We have adjusted this accordingly. See line 721.

- Please correct the author name discrepancy: Rocío Gómez in the ms file versus Rocio Gomez Lencero in our online submission system.

Rocío Gómez is the author's choice of author publication name, which has been used for decades (including publications with EMBO Reports), and they want this to be the name used on the manuscript. However, their legal name is Rocio Gomez Lencero, and this is why their profile included this name. I have deleted "Lencero" from the author profile list online. I hope this is sufficient.

- The author credits need to be removed from the ms file. All credits need to be entered during online ms submission.

We have removed this text.

- The Abbreviations section needs to be removed from the manuscript. Abbreviations should be defined in brackets after their first mention in the text, not in a list of abbreviations.

We have removed the list. All abbreviations are listed after their first mention in the text.

- Disclaimer section and Materials and correspondence should be removed from the ms.

We have removed this text.

- Materials and Methods needs to be called Methods.

We have adjusted this accordingly. See line 588.

- Figure EV4 runs over 3 pages, we don't allow that. All figure must fit onto one page. Please amend.

We have reduced the size of this figure and transferred additional images to the appendix. See Figure EV4 and Appendix Figure S1.

- Please note that the figure 3f-g is mislabeled as figure 3f in the manuscript. This needs to be rectified.

We have adjusted this accordingly. See lines 1137 - 1143.

- Please note that the figure 8b is mislabeled as figure 8d in the data information section of the manuscript. This needs to be rectified.

We have adjusted this accordingly.

- Please note that the exact p values are not provided in the legends of figures 2i-j, l; 3f-g; 4g, i; 5b, f, h, j, n; 6d; 9c-d; EV 1f; EV 3h. Please add the exact values.

We have adjusted this accordingly. See the figure legends of each corresponding figure.

- Please note that the scale bar is missing for figure 9b.

We have made this correction. See Figure 9b.

- Please note that the scale bar needs to be defined for figure EV 3g.

We have made this correction. See lines 1434 - 1435.

I would like to suggest a few minor changes to the abstract that needs to be written in present tense. Please also add information on the species used in this study to the abstract. Please let me know whether you agree with the following:

Centrosomes are the canonical microtubule organizing centers (MTOCs) of most mammalian cells, including spermatocytes. Centrosomes comprise a centriole pair within a structurally ordered and dynamic pericentriolar matrix (PCM). Unlike in mitosis, where centrioles duplicate once per cycle, centrioles undergo two rounds of duplication during spermatogenesis. The first duplication is during early prophase I, and the second is during interkinesis. Using mouse mutants and chemical inhibition, we here block centriole duplication during spermatogenesis and show that non-centrosomal MTOCs (ncMTOCs) can mediate chromosome segregation. This mechanism is different [OK?] from the acentriolar MTOCs that form bipolar spindles in oocytes, which require PCM components, including gamma-tubulin and CEP192. From an in-depth analysis, we identify six microtubule-associated proteins, TPX2, KIF11, NuMA, and CAMSAP1-3, that localize to the non-centrosomal MTOC. These factors contribute to a mechanism that ensures bipolar MTOC formation and chromosome segregation during spermatogenesis when centriole duplication fails. However, despite the successful completion of meiosis and round spermatid formation, centriole inheritance and PLK4 function are required for normal spermiogenesis and flagella assembly, which are critical for fertility.

We have made the adjustments as suggested. See line 25.

EMBO press papers are accompanied online by A) a short (1-2 sentences) summary of the findings and their significance, B) 2-3 bullet points highlighting key results and C) a synopsis image that is exactly 550 pixels wide and 200-600 pixels high (the height is variable). The synopsis image should provide a sketch of the major findings, like a graphical abstract. Please note that text needs to be readable at the final size. Please send us this information along with the final manuscript.

To address A) and B): We have included a document called "EMBOR-2023-58638V3 - Summary statement and bullet points".

To address C): We have also included a synopsis image called "EMBOR-2023-58638V3 - Graphical abstract".

Referee #1:

I stay with my previous evaluation that the work of Skinner et al. is innovative and interesting. Having revised their work, the authors further characterize a centriole-independent pathway in male meiosis that even does not require key centrosomal proteins but runs via CAMSAP, NuMA and TPX2. A main point of criticism was that the negative conclusion of the pathway not requiring otherwise central spindle pole organizers such as Dynein, KIF11, Aurora A and g-TuRC was not well documented. Several approaches added to the manuscript (Fig. 5 and 8 in particular) document the effort to test these activities, or their localization, and either confirm (Dynein, Aurora A, g-TuRC) that they are not involved, or a requirement (KIFF11). Furthermore, the authors have addressed several minor points I had raised (see e.g. Fig. 6B). Taken together, to my opinion, the current version is

significantly increased in clarity & quality to justify publication of the manuscript in EMBO R in its present form.

Thank you!

Referee #2:

The authors have addressed most of the concerns raised by the reviewers by both text changes and addition of new data.

Regarding referencing previous work in other systems (reviewer 3, point 1), the authors have now included a discussion of these studies. However, I strongly suggest to mention and cite some of the key findings from these other systems already in the introduction, to position the current study in the appropriate context. This is important for the non-expert reader to understand the premise for the current work and be able to assess the novelty of the findings.

In the introduction, we have included additional references to studies on other model organisms. See lines 75 - 77.

Dr. Philip Jordan
Johns Hopkins University Bloomberg School of Public Health
Biochemistry and Molecular Biology
615 N. Wolfe St
Room E8626
Baltimore, MD 21205
United States

Dear Dr. Jordan,

I am very pleased to accept your manuscript for publication in the next available issue of EMBO reports. Thank you for your contribution to our journal.
